# Exploring vertical motions in convective and stratiform precipitation using spaceborne radar observations: Insights from EarthCARE and GPM coincidence dataset

Shunsuke Aoki[1], Takuji Kubota[1], F. Joseph Turk [2]

[1]Earth Observation Research Center, Japan Aerospace Exploration Agency, Tsukuba, Ibaraki 305-8505, Japan
[2]Joint Institute for Regional Earth System Science and Engineering, University of California, Los Angeles, CA, USA

*Correspondence to*: Shunsuke Aoki (aoki.shunsuke@jaxa.jp)

**Abstract.**

With the Doppler velocity ($V_d$) measurements from the Cloud Profiling Radar (CPR) onboard the Earth Cloud Aerosol and Radiation Explorer (EarthCARE), it has become possible to observe the vertical motions of hydrometeors inside cloud and precipitation globally. While W-band radar observations by CPR can capture clouds and upper-level ice hydrometeors well, Ku- and Ka-band radar observations by the Dual-frequency Precipitation Radar (DPR) onboard the Global Precipitation Measurement (GPM) Core Observatory are more effective under conditions involving rain or moderate-to-heavy ice precipitation, where attenuation and multiple scattering hinder reliable reflectivity measurements by CPR. This study constructed the EarthCARE–GPM coincidence observation dataset and investigated hydrometeor fall speeds and vertical air motion in stratiform and convective precipitation systems by integrating the complementary information from the two radars. Two case studies were conducted for stratiform and convective events, along with statistical analyses of reflectivity and $V_d$ using nearly one year of dataset. CPR well captured ice particle growth in the upper troposphere above −10°C, while DPR captured the properties of larger hydrometeors in the lower layers, including melting and rain layers. $V_d$ generally increased with decreasing altitude, which is consistent with particle growth inferred from reflectivity observations from both CPR and DPR. Classification into four precipitation types based on echo top heights showed distinct differences in vertical profiles. In deep stratiform cases, $V_d$ reveals slow downward speeds above the melting layer and faster speeds below, consistent with the bright band observed by DPR. $V_d$ in deep convective types indicates faster-falling speed of densely rimed ice particles with high reflectivity and the presence of stronger updrafts and turbulence compared to stratiform cases. These findings indicate that $V_d$ can provide insights into dynamical and microphysical processes inside deep clouds where the quality of reflectivity measurements in W-band deteriorates, and support future development of algorithms for precipitation retrieval and classification using $V_d$.

## 1 Introduction

Precipitation is not only the key component of the global water cycle but also influences atmospheric circulation through the release of latent heat and affects the Earth's radiation budget via their associated cloud distributions. The vertical distribution of diabatic heating varies substantially depending on the type of precipitation system—such as deep convection, stratiform cloud decks, or shallow precipitation—as well as its life cycle. Therefore, understanding the vertical structure and microphysical characteristics in precipitating clouds is essential for gaining insight into the dynamical circulation and the transport of water and radiative energy that govern climate variability. However, our understanding of dynamical and microphysical processes inside clouds remains limited, which contributes to significant uncertainty in weather and climate prediction models.

In numerous past studies, precipitation systems have been classified into convective and stratiform types based on differences in their dynamical and microphysical processes (Houze, 2014). Convective precipitation is characterized by intense, narrow updrafts that promote the formation of large hydrometeors through riming and coalescence, leading to heavy rainfall over limited spatial areas. In contrast, stratiform precipitation typically involves mesoscale upward motion above the melting layer associated with the particle growth through vapor deposition and aggregation, and mesoscale subsidence below the cloud base due to evaporation of rain drops. This type of systems produces weaker but more widespread rainfall compared to convective systems. Our current understanding of vertical air motion in different precipitation types has been developed by synthesizing insights from various observational approaches, with ground-based radars with Doppler velocity capabilities playing a particularly important role. Notable examples include observations using millimeter-wavelength radars such as Ka- or W-band systems (Kollias et al., 2003; Deng et al., 2014), L-band wind profilers or boundary-layer radar (Williams et al., 1995), and VHF atmospheric radars (Mega et al., 2012; Williams, 2012).

However, because the spatial and temporal structures of precipitation systems vary substantially depending on geographic and environmental conditions, satellite observations with global coverage, including over oceans and remote regions, are indispensable for improving our understanding of their connection to the climate system. Among the various satellite-based instruments, spaceborne radar has been particularly valuable for its unique capability to directly observe the vertical structure of clouds and precipitation with high resolution (Battaglia et al., 2020; Nakamura, 2021).

The Tropical Rainfall Measuring Mission (TRMM) satellite was launched in 1997, and it carried the world's first spaceborne Ku-band (13.8 GHz) radar, Precipitation Radar (PR), developed by the Japan Aerospace Exploration Agency (JAXA) and the National Institute of Information and Communications Technology (NICT) (Kummerow et al., 1998; Kozu et al., 2001; Takahashi et al. 2016) . The PR was primarily designed for tropical rainfall observation, and its nearly 17-year mission provided a wealth of data that significantly enhanced our understanding of the vertical structure and diurnal variability of precipitation systems in the tropics (e.g., Nakamura, 2021; Aoki and Shige, 2023). Its successor, the Global Precipitation Measurement (Hou et al., 2014; Skofronick-Jackson et al., 2017) Core Observatory, was launched in 2014. The satellite carries the Dual-frequency Precipitation Radar (DPR). The DPR was developed by the JAXA and the NICT which observes at both

Ku- and Ka-band frequencies (13.6 GHz and 35.55 GHz, respectively) (Kojima et al., 2012; Iguchi 2020). The DPR has continued observations for over a decade, expanding coverage to higher latitudes and offering improved sensitivity (Masaki et al., 2020). It can detect not only rain but also snow, heavy ice particles, and light precipitation, contributing to a deeper understanding of precipitation microphysics (e.g., Yamaji et al. 2020; Seto et al. 2021; Nakamura 2021). In addition to radar observations, multi-frequency microwave radiometers such as the TRMM Microwave Imager (TMI) and the GPM Microwave Imager (GMI), developed by the NASA, with larger spatial coverage have enabled the development of global precipitation datasets, such as GSMaP (Kubota et al. 2020a) and IMERG (Huffman et al. 2020), which provide hourly and half-hourly estimates and have served as a foundation for constructing baseline climatologies of convective and stratiform precipitation.

The Cloud Profiling Radar (Tanelli et al., 2008) onboard CloudSat (Stephens et al., 2002, 2008, 2018), launched in 2006, was the first spaceborne W-band radar (94 GHz) capable of detecting signals from much smaller hydrometeors than those observable with DPR, with a sensitivity down to approximately –30 dBZ. This allows for more accurate quantitative estimates of weakly scattering precipitation, including cloud ice, snowfall, and light rain (Behrangi et al., 2016; Berg et al., 2010; Skofronick-Jackson et al., 2019).

Most recently, the Earth Cloud Aerosol and Radiation Explorer (EarthCARE) satellite (Illingworth et al., 2015; Wehr et al., 2023), launched in May 2024, is equipped with four instruments employing complementary observation techniques: radar, lidar, imager, and radiometer. Among them, the Cloud Profiling Radar (CPR), developed by the JAXA and the NICT, extends the legacy of W-band spaceborne radar observations initiated by CloudSat. Notably, it is the first spaceborne radar in the world to provide Doppler velocity measurements, enabling direct observation of the vertical motion of clouds and precipitation particles.

Before the EarthCARE era, convective-stratiform classification of spaceborne radars relied primarily on spatial features in radar reflectivity fields. For example, (Steiner et al., 1995) proposed a method based on the presence of localized maxima in horizontal reflectivity, while Awaka et al., (1997, 2009) introduced a classification according to the vertical profile of reflectivity that judges the presence or absence of a bright band near the melting layer. Other approaches inferred precipitation growth processes from differences between cloud-top and precipitation-top heights (Masunaga et al., 2005; Takahashi and Luo, 2014), and the GPM mission further incorporated dual-frequency reflectivity ratio into classification algorithms (Awaka et al., 2016, 2021). While these methodologies have offered important insights, they remain partially inferential, as they rely on indirect information derived from reflectivity patterns. EarthCARE's spaceborne Doppler velocity measurements now allow direct observation of vertical cloud motions, offering the potential for a fundamental shift toward process-oriented classification and deeper insights into precipitation dynamics and microphysics on a global scale.

It is important to note, however, that while the W-band CPR offers significant advantages in detecting weakly scattering hydrometeors, it also presents limitations in regions of strong scattering, such as heavy ice and intense rainfall. In these conditions, the signal is affected by attenuation, non-Rayleigh scattering, and multiple scattering effects (Lhermitte, 1990; Meneghini and Kozu, 1990; Battaglia et al., 2011), which complicate the interpretation of the observed data. Moreover, unlike

the PR and DPR, which provide cross-track scanning capabilities, the CPR acquires measurements only in the nadir direction. As a result, it offers limited capability for capturing the horizontal spatial patterns of precipitation systems.

To overcome the limitations inherent to individual satellite sensors, a complementary use of TRMM/GPM and CloudSat observations has been proposed. Some studies combined statistical analyses of precipitation based on radar observations from

GPM and CloudSat to investigate the distribution and structural characteristics of precipitation systems (Hayden and Liu, 2018; Aoki and Shige, 2021). More direct comparisons using coincident observations have also been conducted using CloudSat–GPM coincidence dataset (CSATGPM; Turk et al., 2021). This dataset provides "pseudo three-frequency" radar profiles and coincident passive microwave observations, enabling synergistic analysis of cloud and precipitation properties. In addition, the CloudSat–TRMM coincidence dataset (CSATTRMM; Turk et al., 2021) contains an even larger number of cases

than CSATGPM, as it spans the period prior to CloudSat's transition to Daylight-Only Operations in 2011 (Stephens et al., 2018). Both datasets have been widely utilized in a variety of scientific studies, including investigations into the sensitivity of radar instruments to snow and light rainfall, the structural properties of deep convection, the detection of shallow precipitation, and the development of combined radar–radiometer retrieval algorithms (Arulraj and Barros, 2017; Chase et al., 2022, 2025; Ohara et al. 2025).

In this study, a coincidence dataset combining observations from the EarthCARE/CPR and the GPM/DPR is constructed to facilitate a joint analysis of precipitation systems. Using this dataset, we investigate how Doppler velocity measurements differ between convective and stratiform precipitation regimes. The analysis further examines whether the characteristics of vertical air motion and hydrometeor fall speeds inferred from Doppler observations are consistent with the established understanding of the dynamical and microphysical processes associated with different types of precipitation systems. The GMI is co-located

with DPR and provides the wide swath passive microwave imagery used by global precipitation products such as GSMaP, and EarthCARE data will be useful to evaluate GMI precipitation. However, in this manuscript, only DPR data are presented for comparison with EarthCARE radar profiles.

The remainder of the paper is organized as follows. Section 2 describes the construction of the coincident observation dataset, along with interpretation of Doppler velocity measurements from the CPR. In Section 3, we present case studies and statistical

analyses of cloud and precipitation profiles classified by convective and stratiform types observed by spaceborne radars. We then examine how the observed differences in reflectivity and Doppler velocity among the identified types reflect differences in physical processes, particularly vertical motions such as air updrafts and hydrometeor terminal velocities. Section 4 summarizes the key findings of this study and discusses future tasks.

## 2 Data and methods

### 2.1 EarthCARE – GPM coincidence

EarthCARE/CPR is the W-band radar with the capability of the Doppler velocity measurement. The science data processing chain in the EarthCARE mission was summarized in Eisinger et al. (2024). In this study, EarthCARE L1B CPR one-sensor products (JAXA, 2024a) from 1 August 2024 to 30 June 2025 was utilized, providing radar reflectivity factor and Doppler velocity products. Although the native footprint diameter is about 750 m, the product provides data at 500 m intervals along the track. The vertical resolution is 500 m with 100 m vertical grid spacing. In this study, 5 km along-track integration is applied for each CPR grid point using neighboring 10 grid points to mitigate the effects of footprint differences between CPR and DPR. This horizontal integration also helps reduce the errors that contaminate the Doppler velocity, as described in Section 2.2. The integrated data retains the original 500 m spacing, rather than being resampled at 5 km intervals. The CPR was designed to achieve a sensitivity of approximately –35 dBZ with 10 km integration. The EarthCARE spacecraft operates in a sun-synchronous polar orbit with an inclination angle of 97.05°, crossing over the equator at local times of 02:00 and 14:00. In Section 3.4, the vertical air motion provided by the EarthCARE L2a CPR one-sensor cloud products (CPR_CLP) (Sato et al. 2025) Version Bb (JAXA, 2024b) was utilized, moving-averaged horizontally over 5 km to match the DPR footprint.

To ensure the precision of Doppler velocity measurements, EarthCARE/CPR operates at a higher pulse repetition frequency (PRF) than CloudSat/CPR. As a result, second-trip echoes due to mirror images and multiple scattering more frequently appear at higher altitudes. Following the approach of Battaglia (2021), we estimated the echo power of second-trip returns and applied corresponding masks to remove these artifacts, as described in Aoki et al. (2025). Pre-launch studies suggested that the EarthCARE/CPR would be less affected by surface clutter than the CloudSat/CPR and that, over flat surfaces, clutter would not extend above 600 m (Roh et al., 2023). Observations are consistent with this expectation, although the altitude affected by clutter is higher over mountainous terrain. Taking these factors into account, we excluded data within five range bins (~500 m) above the ocean surface and ten range bins (~1000 m) above land to avoid potential contamination from ground clutter. Interpretations of Doppler velocities are described in Section 2.2.

GPM/DPR consists of two radars: the Ku-band radar (KuPR) and the Ka-band radar (KaPR). Both radars operate with 49 beams in the cross-track direction, providing a swath width of approximately 250 km and a horizontal footprint of about 5 km in diameter (Kojima et al. 2012). The KuPR has a vertical resolution of 250 m and a minimum detectable reflectivity of 15.71 dBZ (Masaki et al. 2020). The KaPR supports two observation modes: High Sensitivity (HS) and Matched Scan (MS). In this study, only data from the HS mode are used in the statistical analysis from Section 3.2 onward, which offers a vertical resolution of 500 m and a minimum sensitivity of 13.71 dBZ (Masaki et al. 2020). The altitude of the GPM Core Observatory was raised from 407 km to 442 km in November 2023 (Kubota et al. 2024). GPM Core Observatory observations that intersect with the EarthCARE satellite are only be available after the altitude change.

We used the GPM/DPR L2 Precipitation (hereafter, referred to as "2A.DPR") (JAXA, 2014) that corresponds to the same observation period as the CPR. In 2A.DPR algorithm, the measured radar reflectivity is corrected for attenuation using the

method of Seto et al. (2021). For the convective-stratiform classification in Section 3, we adopted the "typePrecip" flag provided in the 2A.DPR product (Awaka et al., 2021), which labels each radar footprint as either stratiform, convective, or others. Temperature data were provided by the 2A.DPR product, which were calculated from global analysis data of the Japan Meteorological Agency by interpolating to fit the footprint of the DPR (Kubota et al. 2020b).

The GPM satellite flies in a non-sun-synchronous orbit with an inclination angle of 65°, enabling observations of the low- and mid-latitude regions at a wide range of local times. Because of this unique orbital configuration, GPM ground tracks intersect with those of many sun-synchronous satellites, including EarthCARE, at multiple locations across the globe.

To jointly utilize three-frequency radar reflectivity and W-band Doppler velocity observations, we constructed the EarthCARE – GPM coincidence dataset, following a similar approach to that used for the 2B.CSATGPM dataset (Turk et al., 2021). Note that CloudSat was in Daylight-Only Operations while the GPM Core Observatory was in operation. The EarthCARE-GPM coincidence dataset allows for day and night analysis, which was impossible in the CSATGPM dataset. All cases in which the ground tracks of EarthCARE and GPM intersected within a 15-minute time window were identified over the period from August 2024 to June 2025. For each coincidence event, the nearest DPR footprint was matched to every CPR horizontal grid points with 500 m spacing.

Figure 1 shows the latitudinal distribution of coincident observation footprints between EarthCARE/CPR and GPM/DPR. The number of coincidences increases with latitude and peaks around 65°, corresponding to the edge of the DPR's observation area. While a larger allowable time difference increases the number of samples, it also raises the risk of mismatches due to the movement and evolution of precipitation systems. In this study, a 15-minute threshold is adopted, consistent with the 2B.CSATGPM dataset, to ensure a balance between minimizing observation mismatch and maintaining sufficient number of samples.

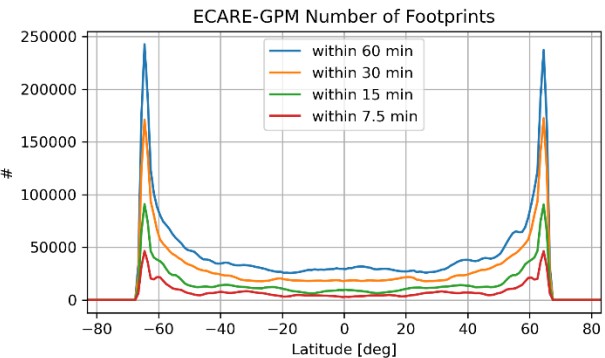

**Figure 1: The number of coincident observation footprints between EarthCARE/CPR and GPM/DPR at each 1-degree latitude interval. Each line indicates the number of footprints where EarthCARE and GPM crossed within 60, 30, 15, and 7.5 minutes, respectively.**

Given the higher sensitivity of CPR compared to both KuPR and KaPR, the echo top height (ETH) observed by CPR is, in principle, expected to be higher. However, in some cases, KuPR or KaPR report higher ETHs than CPR, likely due to temporal mismatches between sensors or spatial mismatches caused by differences in footprint size. In the statistical analyses, such cases were excluded. Additionally, in instances where multiple cloud layers were detected by CPR (e.g., upper-level anvils overlying shallow clouds), only the cloud layer that overlapped with the DPR echo region was extracted for analysis.

## 2.2 Interpretations of Doppler velocities

The Doppler velocity measured by CPR ($V_d$) can be expressed as

$$V_d = V_{air} + V_{tz} + \varepsilon, \tag{1}$$

where $V_{air}$ is the vertical air motion, $V_{tz}$ is the reflectivity-weighted terminal velocity of hydrometeors, and $\varepsilon$ represents the measurement error. In this paper, positive values of $V_d$ are defined as upward direction. Therefore, positive $V_{air}$ means upward air motion and $V_{tz}$ is always negative. The reflectivity-weighted terminal velocity $V_{tz}$ is further formulated as

$$V_{tz} = \frac{\int v_t(D) N(D) \sigma_b(D) \, dD}{\int N(D) \sigma_b(D) \, dD}, \tag{2}$$

where $v_t(D), N(D), \sigma_b(D)$ are the terminal velocity, particle size distribution function, and backscattering cross-section for each particle diameter $D$, respectively. The denominator of Eq. (2), when multiplied by $\lambda^4/\pi^5 |K_w|^2$, where $\lambda$ is the radar wavelength and $K_w$ is the normalizing dielectric factor, corresponds to the radar reflectivity factor ($Z$). The error term $\varepsilon$ includes several components: random noise due to reduced decorrelation arising from the fast-moving speed of the satellite-based sensor ($\varepsilon_{random}$) (Doviak and Zrnic, 1993; Hagihara et al., 2023); uncertainty caused by pointing due to perturbation in satellite attitude and antenna thermal distortion ($\varepsilon_{pointing}$) (Tanelli et al., 2005); uncertainty due to velocity aliasing or folding beyond the Nyquist limit, non-uniform beam filling (NUBF) (Sy et al., 2014), and multiple scattering (Battaglia et al., 2011). In Eq. (2), $\varepsilon$ is expressed as an additive term, but it includes not only systematic biases, but also random uncertainties mentioned above that can only be mitigated by adaptive filtering or along-track integration.

To correct for $\varepsilon_{pointing}$, we applied a bias correction based on the assumption that the 100-km horizontally averaged Doppler velocity at the surface should be zero. In the averaging, only horizontal grid points whose normalized surface cross section ($\sigma^0$) satisfies $-15 < \sigma^0 < 27.5$ dB are used to exclude cases where the surface echo is either too weak due to cloud attenuation or too strong, causing $V_d$ to be severely affected by surface property. This approach compensates for periodic biases of approximately 0 to 0.5 m s$^{-1}$, which are thought to result from thermal distortion of the antenna leading to slight variations in pointing angle with orbital position. Horizontal 5km-integration introduced in this study suppresses $\varepsilon_{random}$ to a level that does not hinder analysis and also reduces the impact of NUBF. To mitigate contamination from multiple scattering and heavily attenuated cases, we followed the method of Battaglia et al. (2011) and excluded range bins where the cumulative reflectivity

from the top of the atmosphere exceeds a specified threshold. Finally, an aliasing correction was applied to $V_d$ following the method of Hagihara et al. (2023) and was further refined using the surrounding $Z$ and $V_d$ profiles as a reference.

Previous studies using various types of radar have attempted to derive $V_{air}$ by subtracting the estimated $V_{tz}$ from the observed Doppler velocity (Kollias et al., 2003; Okamoto et al., 2003; Yamamoto et al. 2008; Sato et al. 2009, 2010). Their methodology provides $V_{air}$ by estimating the particle type and size distribution from the reflectivity profile and path-integrated attenuation, and then subtracting the corresponding $V_{tz}$. This may yield reasonable estimates of $V_{air}$ in ice particle regions. However, in rain regions where attenuation is strong, it is difficult to accurately correct radar reflectivity for attenuation, which makes the retrieval of the particle size distribution challenging.

In contrast, $V_d$ is retrieved from the pulse-to-pulse phase difference rather than from signal amplitude (Eisinger et al. 2024), and is therefore intrinsically less affected by attenuation of returned power (Doviak and Zrnic, 1993). In practice, pulse-to-pulse phase correlation is maintained under moderate attenuation, allowing the velocity retrievals to remain stable (Tian et al., 2007; Kollias et al., 2014). The main limitation arises when severe attenuation and multiple scattering associated with heavy rain or ice precipitation substantially degrades the pulse-to-pulse phase correlation, leading to large errors (Matrosov, 2008; Battaglia et al., 2011). In this study, cases containing rain, wet snow, and graupel were retained, while severe attenuation were excluded by applying screening following Battaglia et al. (2011), leading to preserve physically meaningful velocity information in these hydrometeor regimes.

In the current study, we propose a method for estimating $V_{air}$ by subtracting the reflectivity-weighted terminal fall velocity $V_{tz}$ which is estimated from the particle size distribution derived from DPR measurements, from the observed $V_d$ from collocated CPR measurements. In 2A.DPR algorithm (Seto et al., 2021), drop size distribution function is expressed as

$$N(D) = N_w f(D; D_m) = N_w \frac{6(\mu+4)^{\mu+4}}{4^4 \Gamma(\mu+4)} \left(\frac{D}{D_m}\right)^\mu \exp\left(\frac{(\mu+4)D}{D_m}\right), \tag{3}$$

where $\Gamma$ is the Gamma function with the value of $\mu$ fixed at 3, $N_w$ is the parameter related to the number concentration of raindrops, and $D_m$ is the mass-weighted mean diameter. Substituting Eq. (3) into Eq. (2) yields

$$V_{tz} = \frac{\int v_t(D) f(D; D_m) \sigma_b(D)\, dD}{\int f(D; D_m) \sigma_b(D)\, dD}. \tag{4}$$

From this formulation, $V_{tz}$ can be uniquely determined from the DPR-estimated $D_m$, and thus $V_{air}$ can be calculated. For rain layers, $v_t$ was computed using the empirical relationship proposed in Atlas and Ulbrich (1977), with a correction factor for air density, as given:

$$v_t(D) = -3.78 D^{0.67} \cdot c(\rho), \tag{5}$$

$$c(\rho) = \sqrt{\frac{\rho_0}{\rho}} = \sqrt{\frac{\rho_0 RT}{p}}. \tag{6}$$

Here, the unit of $D$ is millimeters, $\rho$ denotes the ambient air density, $\rho_0$ is the standard air density (set to 1.225 kg m⁻³), $R$ is the specific gas constant for dry air (287 J kg⁻¹ K⁻¹), and $p$ and $T$ represent pressure and temperature obtained from auxiliary

data. The backscattering cross-section $\sigma_b$ was derived from Mie scattering calculations for spherical raindrops at W-band frequency.

For snow, $\sigma_b$ and $v_t$ was calculated in the same manner as in the 2A.DPR algorithm, assuming homogeneous spherical particles with a density of 0.10–0.13 g cm⁻³ and a melted-equivalent diameter following the particle size distribution given by Eq. (3). The terminal fall velocity of snow was calculated following Magono and Nakamura (1965) as follows:

$$v_t(D_s) = -8.8(0.1D_s\rho_s)^{0.5} \cdot c(\rho), \tag{7}$$

where $D_s$ is the unmelted snow particle diameter in mm, and $\rho_s$ is the density of snow particles in g cm⁻³. On the other hand, ice particles can take various shapes, sizes, and densities, such as those of snow, graupel, and hail. Because $\sigma_b$ and $v_t$ vary depending on these parameters, the assumptions made for snow in this study are often not valid. Although it would be ideal to account for more realistic and complex scattering and fall characteristics of ice particles (Kuo et al. 2016; Ori et al. 2021), considering such diversity is challenging because the CPR observes only in the nadir direction and therefore cannot provide information on particle asymmetry. This contrasts with ground-based dual-polarization radars, which observe the hydrometeor from the side, where particle asymmetry is more evident and can provide additional information. In addition, such information on particle diversity cannot be inferred from the current version of the 2A.DPR algorithm and is therefore left for future work.

## 3 Results

### 3.1 Case study of CPR and DPR coincidence observations

In this subsection, two cases of CPR and DPR coincidence observations are presented: one in which stratiform precipitation was dominant and one in which convective precipitation was dominant. Figure 2 shows a case over the northeastern Pacific associated with a tropical cyclone observed on 22 August 2024. The EarthCARE and GPM passed over a broad stratiform precipitation system with time difference of approximately 6 minutes (Figs. 2a and 2b). In Fig.2c, the vertical cross-section of W-band radar reflectivity observed by CPR ($Z_W$) reveals deep cloud structures reaching altitudes of approximately 15 km, and detailed texture within the upper-level ice and snow clouds above the 0°C level is evident. Locally enhanced reflectivity just below the 0°C level corresponding to the melting layer is observed. Below the melting level, significant attenuation occurs in the rain region, resulting in reduced $Z_W$ values. The $V_d$ profile clearly distinguishes between the slowly falling ice and snow layers aloft and the faster-falling rain below (Fig. 2d). Because $V_d$ is derived from phase information of the radar signal, it remains observable even in layers affected by attenuation.

Ku-band radar reflectivity retrieved by KuPR ($Z_{Ku}$) shows ETHs around 6 to 10 km, capturing signals only in areas where $Z_W$ exceeds approximately 10 dBZ in the ice phase (Fig. 2e). However, strong scattering from developed snow near the melting layer and rain below is captured effectively without attenuation. A prominent bright band, commonly observed in stratiform

precipitation, is evident near the 0°C level. Ka-band radar reflectivity ($Z_{Ka}$) exhibits a similar profile to $Z_{Ku}$, although the
reflectivity values are lower due to Mie scattering effects and stronger attenuation at Ka-band frequencies (Fig. 2f).

Figure 3 presents a case of scattered convective precipitation observed over southern South America on 23 March 2025, with a time difference of approximately 4 minutes between the EarthCARE and GPM observations. In this event, ETHs and reflectivity maxima observed by CPR and KuPR vary across individual convective cells, and no distinct bright band signature is evident (Figs. 3c–3f). The $V_d$ exhibits considerable variability, with no clear transition near the 0°C level (Fig. 3d). In some
locations, strong scattering in the ice phase causes severe signal attenuation in $Z_W$ measurement, leading to a complete loss of surface returns (e.g., near 30°S). In these areas, regions of high $Z_{Ku}$ (> 30 dBZ) and large downward $V_d$ exceeding 2 m s$^{-1}$ extend above the 0°C level, suggesting the presence of densely rimed particles such as graupel or hail. In contrast, shallow precipitation is characterized by weak reflectivity and small $V_d$ values, indicating a dominance of numerous small droplets, as typically seen in drizzle.

The horizontal 5 km integration applied to the $V_d$ field in Fig. 2d is highly effective in reducing the contribution of random noise induced by decorrelation ($\varepsilon_{random}$). However, in the convective case shown in Fig. 3d, this integration may result in over-smoothing, mixing the signatures of strong echoes within convective cores with weaker echoes at cloud edges. Because the integration is performed as a reflectivity-weighted average rather than a simple moving average, the features of the convective core are emphasized, which may in turn lead to artificially enhanced downward velocities near cloud edges where
echoes are weak. In the statistical analyses presented later in this paper, such edge regions, where coincidence with DPR observations could not be ensured after averaging, were excluded from the analysis, as indicated by the black-plotted areas in Figs. 2g and 3g.

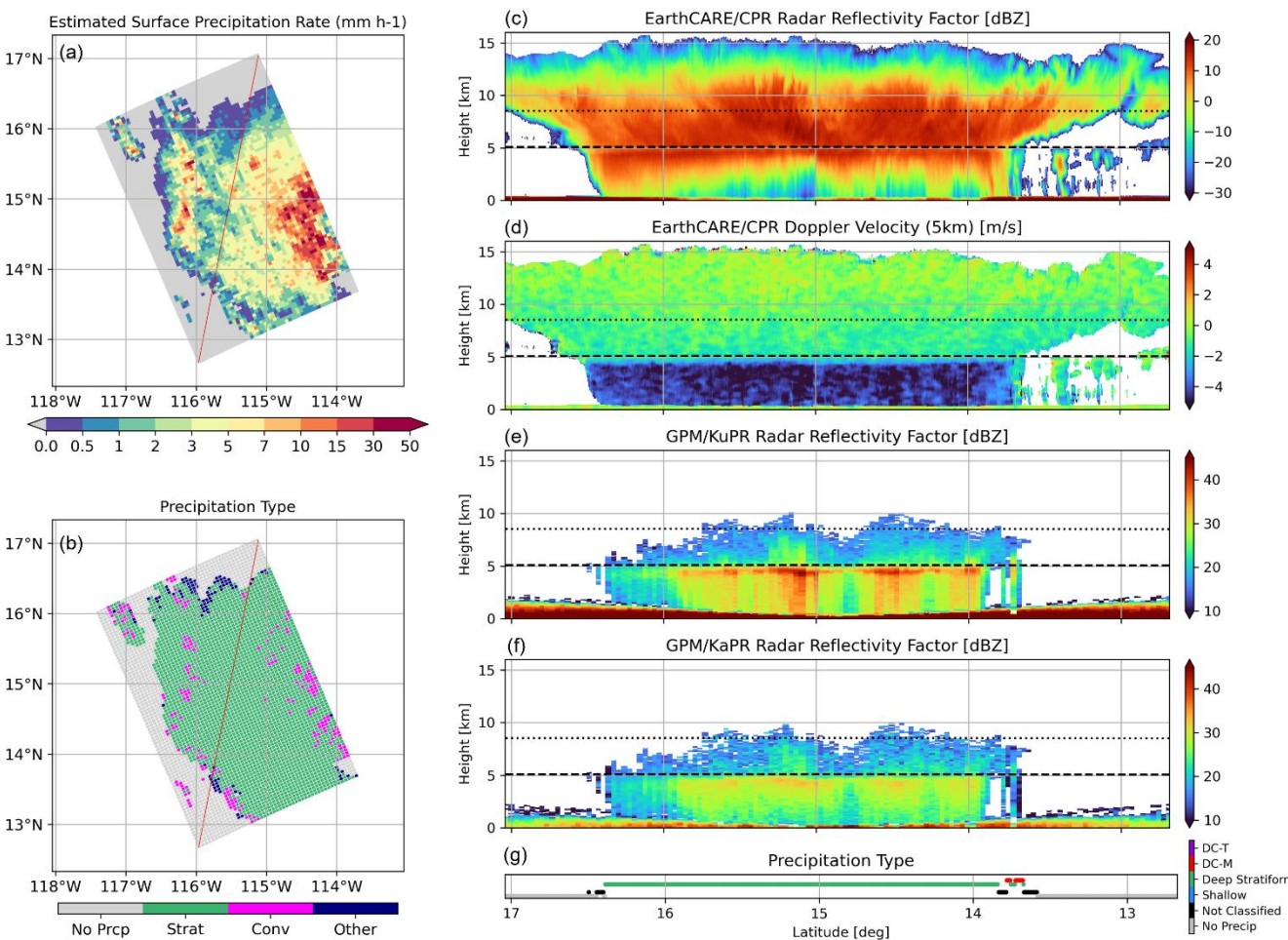

Figure 2: A stratiform precipitation case of the EarthCARE-GPM coincidence. This case is a tropical cyclone over the eastern Pacific around 2150 UTC on 22 August 2024 (frame 1337E). Time difference between EarthCARE and GPM observations is 5.6 minutes. (a) map of estimated surface precipitation rate, and (b) stratiform-convective type classification in 2A.DPR algorithm. The red line indicates the ground track of CPR footprints. Distance-height cross-section of (c) radar reflectivity and (d) doppler velocity integrated over 5km, both measured by CPR. Radar reflectivity from (e) KuPR and (f) KaPR along CPR track. Dashed and dotted lines are the 0 °C and −20 °C isothermal lines, respectively. (g) Precipitation type classification defined in Section 3.3 for each CPR footprint: tall deep convective (DC-T; purple), moderate deep convective (DC-M; red), deep stratiform (green), shallow (light blue), other type of precipitation or unclassified due to data quality issues (black), and no precipitation (grey).

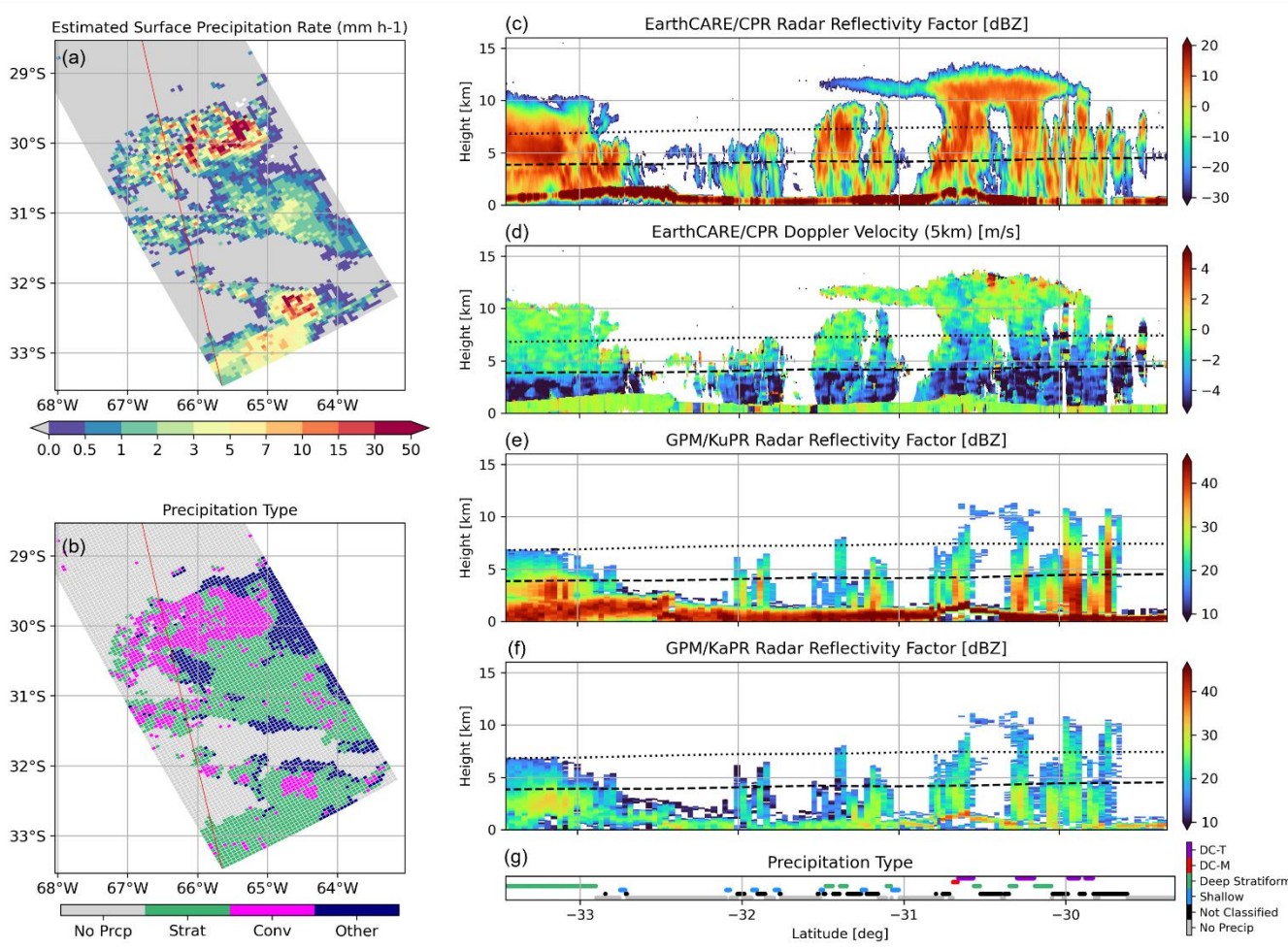

Figure 3: Same as Fig.2, but for scattered convective case over Argentina around 0640 UTC on 23 March 2025 (frame 4641H). Time difference between EarthCARE and GPM observations is 3.8 minutes.

## 3.2 Statistics of Z and Vd profiles with temperature

In this section, we perform a statistical analysis of the coincident profiles of $Z$ and $V_d$ observed by CPR and DPR to examine the characteristics of clouds and precipitation profiles with temperature. Figure 4 presents Contoured Frequency by tEmperature Diagrams (CFEDs; Hashino et al. 2013) of $Z_W$, $V_d$, $Z_{Ku}$, and $Z_{Ka}$, where temperature is used as the vertical coordinate. The probability density function is normalized within each temperature bin. Only profiles where echoes are detected by both DPR and CPR are included. These profiles correspond to 5.1% of all profiles and 10.1% of the profiles in which echoes are detected by the CPR. Because the sensitivity of the DPR is lower than that of the CPR, the number of samples in the colder temperature range in Figs. 4c and 4d is considerably smaller than that for the CPR. To avoid misinterpretation,

temperature ranges with fewer than 1000 samples are not shown. Using temperature instead of altitude as the reference axis allows for more direct interpretation in terms of cloud microphysical processes. To ensure a sufficient number of samples, observations from various regions and times were aggregated over the analysis period. Consequently, the results may include observations from different altitudes depending on latitude and season; however, such variations are beyond the scope of this study.

In Fig. 4, above the freezing level, $Z$ in all frequency bands (W, Ku, and Ka) generally increases with decreasing altitude, and $V_d$ also increases in the downward direction. This indicates that ice particles grow and become dominated by larger sizes, resulting in an increase in their terminal fall speeds. Reflecting the scattering properties specific to each frequency band, the height ranges over which $Z$ increases differ between the W-band, Ku-, and Ka-band observations. As shown in Fig. 4a, above approximately $-20°C$, $Z_W$ increases with decreasing altitude, indicating particle growth with height. However, below $-20°C$, where $Z_W$ exceeds 0 dBZ and the scattering regime likely attenuation and transitions from Rayleigh scattering to Mie scattering, the rate of increase in $Z_W$ becomes smaller. In the height range between $-10°C$ and $0°C$, the $Z_W$ PDF shows little dependence on height, indicating it difficult to detect altitude-dependent variations in microphysical properties from $Z_W$.

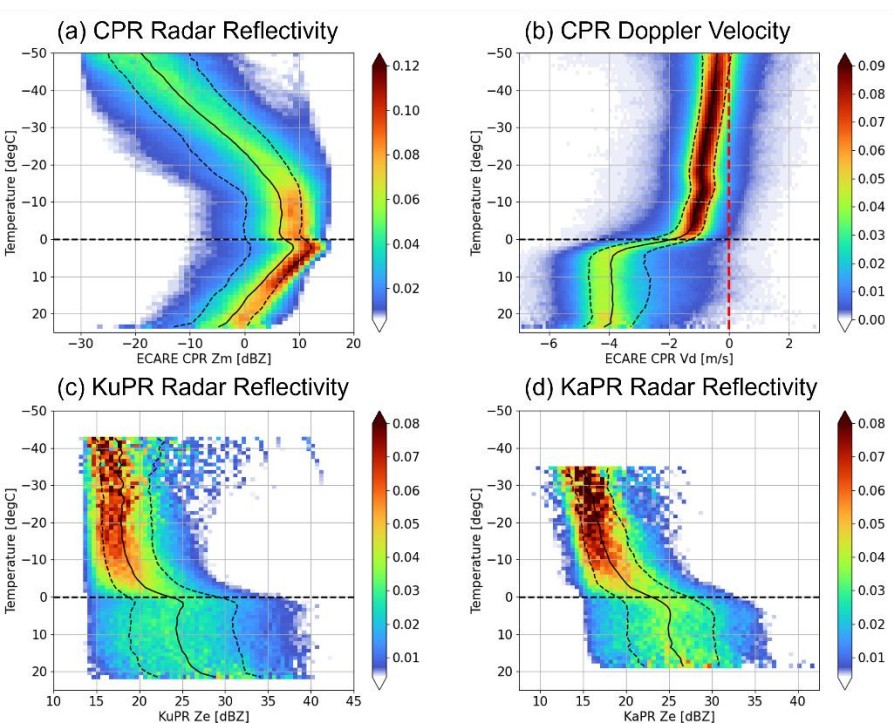

**Figure 4: CFEDs for (a) radar reflectivity and (b) Doppler velocity measured by CPR, and attenuation-corrected radar reflectivity from (c) KuPR and (d) KaPR, constructed using the EarthCARE–GPM coincidence dataset from August 2024 to June 2025. The probability density function is normalized within each temperature bin. Total number of samples in each temperature bin smaller than 1000 are excluded. Black solid lines indicate the median value at each temperature, and dashed lines indicate 20 and 80 percentile values.**

In KuPR and KaPR observations (Figs. 4c and 4d), reflectivity values above −10°C are close to the minimum detectable thresholds of 15 dBZ and 13 dBZ, respectively. Echoes are detected only when hydrometeors with strong reflectivity observed by CPR extend into the upper layers. According to the conversion table in Skofronick-Jackson et al. (2019), this reflectivity level corresponds to approximately 11 dBZ in the W-band. In the layer between −10°C and 0°C, $Z_{Ku}$ and $Z_{Ka}$ increases with decreasing altitude, indicating growth of hydrometeors such as snowflakes or graupel. This vertical evolution, which is not captured in $Z_W$ due to Mie effects and attenuation, is better resolved in the Ku- and Ka-band observations.

On the other hand, the CFED of $V_d$ shown in Fig. 4b illustrates an increase in downward velocity with increasing temperature between −20°C and 0°C, which is consistent with the particle growth inferred from KuPR observations. This suggests that Doppler velocity measurements can provide insights into the development of intense snowfall that could not be captured by CloudSat's reflectivity observations alone.

Around the melting level, a marked change in $V_d$ is observed (Fig. 4b). The downward velocity increases from approximately −1 m s⁻¹ to −4 m s⁻¹, corresponding to the transition from snow to rain. This is accompanied by a localized enhancement in $Z$ of all band, likely due to the contrast in complex refractive indices between ice and water (Figs. 4a, 4c, and 4d). Below the melting level, where rain becomes dominant, $Z_W$ decreases with increasing temperature, primarily due to strong attenuation at W-band frequencies. In contrast, median $V_d$ stays around −4 m s⁻¹ at temperatures above 5 °C, which is consistent with the terminal fall speed of the raindrops. This indicates that $V_d$ is less affected by attenuation than $Z_W$, and it can reflect the properties of hydrometeors even in the rain layer. Meanwhile, Ku-band observations are subject to less attenuation than W-band, enabling more reliable attenuation correction and making $Z$ in the rain layer more representative of the actual precipitation structure.

It should be noted that although $Z_{Ku}$ and $Z_{Ka}$ increase with increasing temperature between 10 °C and 25 °C (Figs. 4c and 4d), this does not necessarily imply stronger precipitation lower in the column. Near the melting layer, the data include contributions from all latitudes, whereas the observations around 20 °C are dominated by low-latitude regions. Although the limited number of samples makes detailed discussion difficult at present, future work should involve analysis separated by weather systems and freezing level height to better investigate the vertical growth processes of hydrometeors.

To examine the relationship between $Z$ and $V_d$ in more detail across different frequencies, histograms were constructed for various temperature ranges, as shown in Fig. 5. The dashed and dotted black lines in Fig. 5 represent the theoretical $Z$–$V_{tz}$ relationship for rain and snow, respectively, calculated under the assumptions described in Section 2.2. Figures 5a–5d use $Z_W$ as the horizontal axis, while Figs. 5e–5h use $Z_{Ku}$. For snow, lines corresponding to $\rho_s$ of 0.05, 0.1, 0.2, and 0.3 g cm⁻³ are plotted. Such $Z$–$V_d$ relationships have long been investigated using ground-based radar observations and serves as a useful metric for inclusion in weather and climate models. In the upper troposphere ($T < -10$°C; Fig. 5a), $V_d$ tends to increase with increasing $Z_W$, indicating that larger reflectivity is associated with faster-falling particles. Assuming that the vertical air motion averages to zero over many samples, the mean $V_d$ can be interpreted as representative of the $V_{tz}$. The $Z_W$–$V_d$ distribution in

Fig. 5a follows the theoretical $V_{tz}$ curve for $\rho_s = 0.05$ g cm$^{-3}$, showing that the downward fall speed increases with increasing $Z_W$, which provides insight into the growth of ice particles. On the other hand, in KuPR observations (Fig. 5e), the distribution is concentrated around 15–18 dBZ, near the instrument's sensitivity limit, suggesting that only stronger signals are detected in these layers due to limited sensitivity.

In the temperature range between −10°C and 0°C, the distribution of $Z_W$ is tightly concentrated around 8–11 dBZ, making it difficult to discern any clear relationship between $Z$ and $V_d$ (Figs. 5b–5d). Moreover, in the melting and rain layers ($T > 0$°C), attenuation becomes significant, making it unclear whether low $Z$ values are due to inherently weak reflectivity or attenuation effects (Figs. 5c and 5d). In contrast, when focusing on $Z_{Ku}$, a negative relationship between $Z_{Ku}$ and $V_d$ is observed (Figs 5f–5h). In the range of −10°C < T < 0°C (Fig. 5f), the distribution closely follows the theoretical curves for snow with densities of 0.05–0.1 g cm$^{-3}$, while in the range of T > 4°C (Fig. 5h), it aligns well with the theoretical curve for rain. In theory, the slope of $Z_{Ku}$ versus $V_{tz}$ is steeper for rain than for snow ($\rho_s \sim 0.1$ g cm$^{-3}$) because the fall velocity of snow exhibits a weaker dependence on particle size compared to that of rain. This trend is also evident in the observations: the $Z_{Ku}$–$V_d$ slope is −0.0170 between −10°C and 0°C, −0.0320 between 0°C and 4°C, and −0.0444 for $T > 4$°C, indicating a steepening slope as precipitation transitions from snow to rain through melting (Figs. 5f–5h). In addition, in the 0–4°C melting layer, $V_d$ exhibits a broad range of values because this region captures the transition from slow-falling particles with terminal velocities around 1 m s$^{-1}$ to faster-falling raindrops with velocities approaching 4 m s$^{-1}$ (Fig. 5g). This suggests that $V_d$ may serve as a useful complementary indicator for estimating precipitation intensity in melting and rain layers by using CPR observations where $Z_W$ alone may be insufficient due to attenuation.

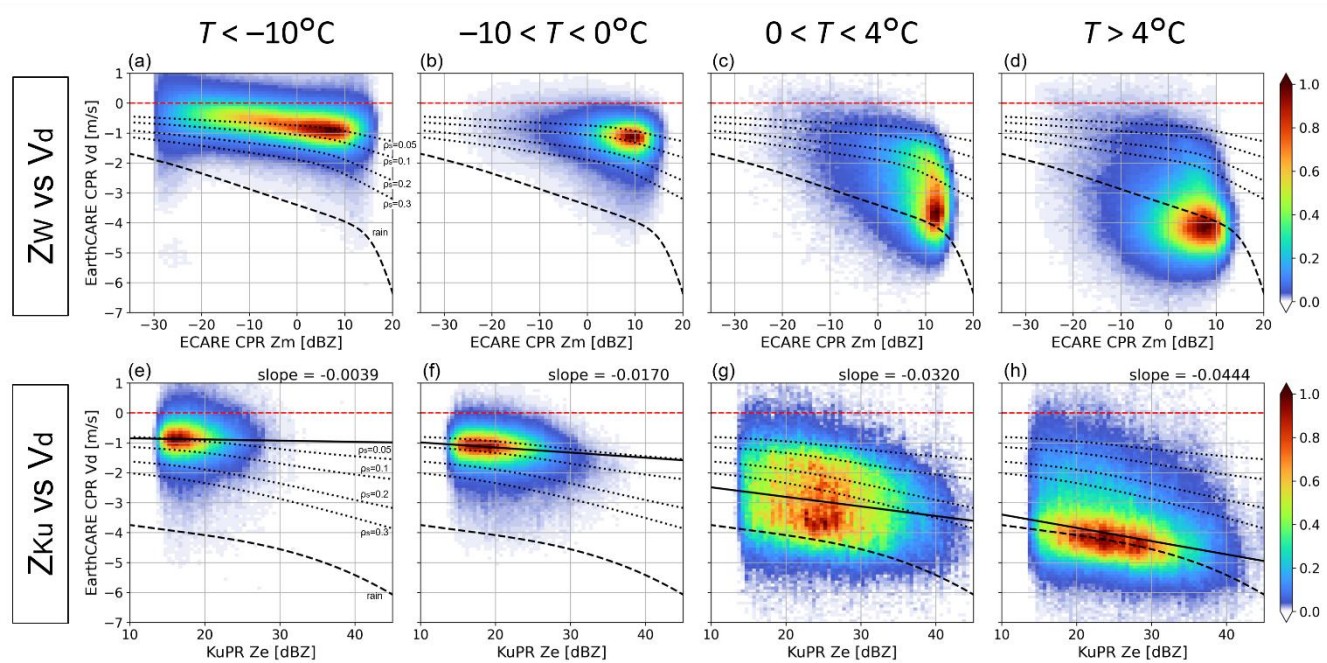

**Figure 5: Joint histograms of radar reflectivity and Doppler velocity for four temperature ranges: (a, e) < −10°C, (b, f) −10 to 0°C, (c, g) 0 to 4°C, and (d, h) > 4°C. Panels (a–d) use CPR radar reflectivity on the x-axis, while (e–h) use KuPR attenuation-corrected radar reflectivity. The red dashed line indicates 0 m/s Doppler velocity. The solid black lines in (e–h) represent regression lines fitted using the least squares method, with their corresponding slopes indicated in the upper right corner outside each panel. The dashed and dotted black lines represent the theoretical $Z-V_{tz}$ relationship for rain drops at 850 hPa and 10°C, and for snowflakes with a densities of 0.05, 0.1, 0.2 and 0.3 g cm⁻³ at 600 hPa and −10°C calculated from eq. (4), where radar reflectivity in W-band and Ku-band is calculated assuming $N_w = 10^3$ mm⁻¹ m⁻³.**

## 3.3 Cloud & precipitation type classification

To investigate how in-cloud statistics of $Z$ and $V_d$ differ according to the characteristics of precipitation systems, a classification of coincident events was conducted, followed by a statistical analysis of the corresponding vertical profiles. The cloud top height (CTH) and precipitation top height (PTH) are key variables that characterize the developmental stage of precipitation systems (Masunaga et al., 2005, hereafter M05; Stephens and Wood 2007, hereafter SW07; Takahashi and Luo, 2014; Kikuchi and Suzuki, 2018). M05 first categorized precipitation systems using CTH–PTH joint histograms constructed by deriving the PTH from the 18-dBZ echo top observed by the TRMM PR and the CTH from the 11-μm brightness temperature observed by the Visible Infrared Scanner (VIRS) onboard TRMM. SW07 improved upon this approach by incorporating millimeter-wavelength radar observations, which allowed them to better represent multilayer cloud structures, whereas VIRS observations can capture only the uppermost cloud layer. Following these studies, this work assumes that the ETH retrieved from CPR corresponds to the CTH, and that from KuPR corresponds to the PTH. The joint histograms of temperature at CTH and temperature at PTH for stratiform and convective precipitation determined by 2A.DPR algorithm,

respectively, are shown in Fig. 6. The histograms are reminiscent of the histograms presented by M05 (their Fig. 1) and SW07
(their Fig. 8).

When the PTH is located above the 0°C level, it is indicative of a cold-type precipitation process, in which ice particles grow into relatively large snow aggregates or graupel through aggregation and riming. In stratiform precipitation, the PTH is mostly at temperatures lower than 0°C, indicating that most cases are associated with cold-type precipitation (Fig. 6a). In particular, a high frequency of occurrence is confined within the PTH range of −20°C to −10°C regardless of CTH. This temperature

range corresponds to the layer where ice habits transition with temperature, as shown in the classical Nakaya diagram (Libbrecht 2005). It is also referred to as the dendritic growth layer, where cloud ice particles are thought to grow into snow through depositional growth, aggregation, and potentially secondary ice processes (von Terzi et al. 2022). Such temperature-dependent microphysical processes may explain why ice particles in typical stratiform clouds become large enough to be detected by the KuPR only when they reach below the −20°C level.

In contrast, convective precipitation shown in Fig. 6b exhibits a much wider range of PTH values, extending from 20°C down to below −40°C, with a sparse distribution in the CTH–PTH histogram. Focusing on deep clouds with CTH above the −20°C level, the PTH tends to lie close to the one-to-one line between PTH and CTH, indicating that the precipitation top height is nearly as high as the cloud top. This situation can be interpreted as the result of strong updrafts within the system that lifted large hydrometeors toward the cloud top, as discussed in Takahashi and Luo (2014). In addition, in convective cases, a

pronounced peak in occurrence is found where the PTH is below the 0°C level. When the PTH is below the 0°C level, it suggests a warm-type precipitation process, where raindrops grow through collision and coalescence of liquid water droplets. Such warm-type shallow precipitation is likely associated mainly with shallow cumulus or congestus clouds, because the DPR has limited sensitivity to detect light precipitation (~1 mm h⁻¹ or less; Hayden and Liu, 2018), and thus does not effectively capture shallow stratus or stratocumulus.

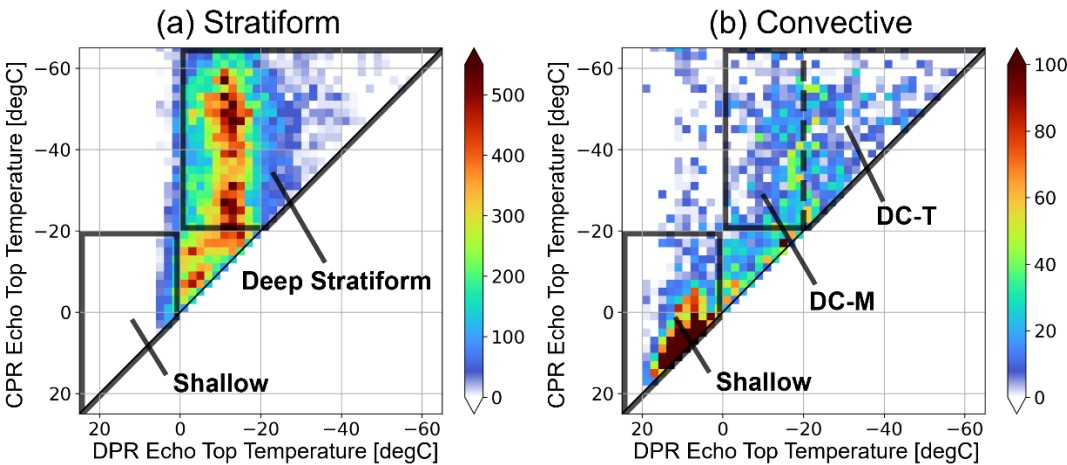

**Figure 6: Joint histograms of the temperature at the echo top height measured by KuPR and the temperature at the echo top height measured by CPR for (a) stratiform and (b) convective precipitation. The black boxes indicate the classification boundaries for the four precipitation types defined in Section 3.3.**

Based on the considerations above, we performed a classification of cloud and precipitation types. Cases where the PTH is below the 0°C level and the CTH is also below the −20°C level were categorized as *Shallow*. This type is intended to extract conditions dominated by warm-type rain processes and includes both convective and stratiform precipitation. Cases with the PTH above the 0°C level and the CTH is also above the −20°C level were classified as *Deep*. In these cases, the cloud layer

extends well above the freezing level, indicative of dominant cold-type processes. Furthermore, the *Deep Convective* category was subdivided into two subtypes: *Moderate (DC-M)* and *Tall (DC-T)*. The *DC-T* subtype represents highly developed convective clouds in which large ice particles are transported to higher altitudes and are detectable by KuPR. In contrast, the *DC-M* subtype likely corresponds to less mature or dissipating convective systems.

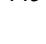

**Figure 7: CFEDs for (a–d) $Z_W$ and (e–h) $V_d$, and (i–l) $Z_{Ku}$ for four precipitation types: Deep stratiform, DC-M, DC-T, and shallow. Black solid lines indicate the median value at each temperature, and dashed lines indicate 20 and 80 percentile values.**

Figure 7 presents CFEDs of $Z_W$, $V_d$, and $Z_{Ku}$ for four precipitation types: Deep Stratiform, DC-M, DC-T, and Shallow. Since the number of stratiform events is substantially larger than that of convective or shallow events, the stratiform CFED in Fig. 7 closely resembles that in Fig. 4. In terms of reflectivity, a defining feature of stratiform precipitation is the presence of a local peak near the melting level, representing a bright band (BB), which is clearly observed in Ku band. In contrast, such features are absent in convective and shallow types. This result is consistent with the classification logic used in the DPR algorithm, in

which the presence of a BB is a key criterion for identifying stratiform precipitation. In convective and shallow types, $Z_{Ku}$ tends to increase with temperature below the melting level without exhibiting a local minimum. The difference from stratiform suggests the growth of raindrops through warm rain processes such as collision and coalescence in these types.

For all three variables, convective precipitation exhibits a broader distribution than stratiform, indicating greater variability in the microphysical properties of falling hydrometeors, more heterogeneous spatial distribution, and more vigorous vertical

turbulent motions. This variability is particularly pronounced in the DC-T category, where fluctuations are especially large.

In Shallow cases, $Z_W$ and $Z_{Ku}$ are generally low, and $V_d$ values are also smaller than in Deep cases. This is consistent with previous studies such as Dolan et al. (2018), based on disdrometer measurements, which indicate a dominance of numerous small raindrops in such conditions in shallow warm-type rainfall.

SW07 has classified cloud types using a similar approach based on ground-based Ka-band radar observations and presented

comparable histograms of $Z$ as a function of height (their Fig. 10). Although the attenuation conditions differ—since their observations are made from the ground upward, whereas the present study is based on spaceborne downward-looking radar measurements—the vertical profiles of $Z_W$ in each category (Fig. 7a–d) exhibit similar characteristics. This study extends their findings by introducing an additional perspective through the use of $V_d$.

**3.4 Discussions on microphysical properties and $V_{air}$**

This section examines how the differences in $Z$ and $V_d$ among precipitation types identified in Section 3.3 can be interpreted in terms of the underlying physical processes in convective and stratiform systems. The analysis focuses on vertical structure, particularly the contributions from vertical air motion and terminal fall velocity. Figure 8 presents the mean and standard deviation of $Z$ and $V_d$ at each temperature for all precipitation types, as derived from the CFEDs in Fig. 7. The discussion is

organized by characteristic layers within the cloud system, namely the snow and rain layers, with particular attention to the $Z$–$V_d$ relationship.

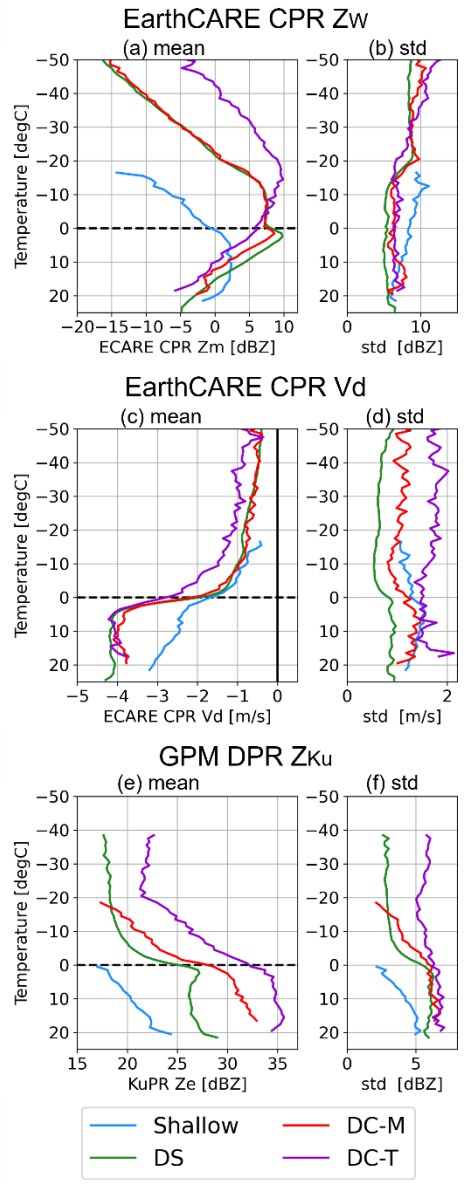

**Figure 8: Vertical profiles of the (a, c, e) mean and (b, d, f) standard deviation of (a, b) $Z_W$, (c, d) $V_d$, and (e,f) $Z_{Ku}$ as a function of temperature for the four precipitation types: Deep stratiform (green), DC-M (red), DC-T (purple), and shallow (light blue).**

### 3.4.1 Snow and ice layers

Above the 0°C level, $Z_W$ in the DC-T type reaches values near 0 dBZ even at temperatures between −30°C and −50°C, which corresponds to more than 5 km above the melting layer (Fig. 8a). This implies that dense ice particles, generated by strong

convective updrafts, are transported to high altitudes. In DC-T profiles, $Z_W$ increases from the cloud top down to around −20°C. In the −20°C to 0°C layer, it saturates and decreases due to attenuation or Mie scattering effects, reaching levels comparable to those in deep stratiform profiles. By contrast, $Z_{Ku}$ remains more than 5 dB higher than in stratiform cases and continues to increase downward in this temperature range (Fig. 8e). This suggests continued growth of snow and ice particles toward larger sizes. The corresponding $V_d$ values in DC-T are also about 1.5 times greater than those in stratiform precipitation, supporting the presence of riming processes that produce dense particles such as graupel and hail (Fig. 8c). Furthermore, the larger standard deviation of $V_d$ compared to that in stratiform precipitation suggests more turbulent vertical air motion, strong wind shear and greater microphysical variability.

DC-M profiles resemble those of stratiform precipitation more closely. $Z_W$ increases between the cloud top and around −20°C and then saturates below. On the other hand, in the $Z_W$-saturated layer between 0°C and −10°C, $Z_{Ku}$ values are 1–2 dB higher than in stratiform cases, indicating ongoing particle growth that is not captured by $Z_W$ alone. The mean $V_d$ is comparable to that of stratiform precipitation, however, its larger standard deviation suggests more active turbulence.

Figures 9a–9c present joint histograms of $Z_{Ku}$ and $V_d$ within the –10°C to 0°C temperature range for deep stratiform, DC-M, and DC-T precipitation types. In the deep stratiform case, the distribution is narrowly concentrated around $V_d \approx −1$ m s⁻¹, consistent with the prevalence of lightly rimed snow particles and relatively weak vertical motions. In Fig. 9c, DC-T shows a broader distribution that extends toward higher $Z_{Ku}$ and faster $V_d$ in downward. This indicates the presence of dense ice particles with larger terminal velocities, likely formed through intense riming processes within strong convective updrafts. The broader spread of the distribution further suggests enhanced turbulence and variability in particle fall speeds.

DC-M in Fig. 9b exhibits characteristics intermediate between stratiform and DC-T, as also indicated by the slope of the regression line. While it shares similarities with deep stratiform in terms of overall structure, it displays a higher occurrence of larger $Z_{Ku}$ and $V_d$ values, along with greater variability. These features imply the presence of moderately rimed particles and more active turbulent mixing than in the stratiform case, although less intense than in DC-T.

In some previous studies, the dual-frequency reflectivity ratio (DFR) has been used to characterize ice-phase precipitation (Leinonen et al., 2015; Yin et al. 2017; Akiyama et al. 2025). Compared with using single-frequency $Z$, DFR cancels the uncertainty associated with the number concentration $N_W$, thereby is more directly related to the particle size distribution and attenuation. Figures 9d–9f show joint histograms with the Ku–Ka band DFR ($Z_{Ku}/Z_{Ka}$) from the DPR plotted on the x-axis. Here, we used $Z$ which was not corrected for attenuation. As in the discussion using $Z_{Ku}$, the convective type tends to show larger DFR values and faster downward $V_d$ compared to the stratiform type, with a steeper regression slope in the DFR–$V_d$ relationship. This suggests the dominance of larger particles with higher density. However, correlation coefficients for each case (Table 1) show that $Z_{Ku}$ correlates more strongly with $V_d$ than DFR does. This simple analysis therefore does not demonstrate a clear advantage of using DFR. The spread of $V_d$ with respect to DFR may reflect the variations in microphysical characteristics such as particle shape and density, as well as atmospheric turbulence. Moreover, because the DFR was calculated using the KaPR HS observation swath, the number of samples is about half that of Figs. 9a–9c, which may have

resulted in the lower correlation. As future work, once a larger multi-year dataset becomes available, scattering calculations that account for variations in ice particle shape and density will enable DFR to provide more detailed insights into cloud and precipitation microphysics.

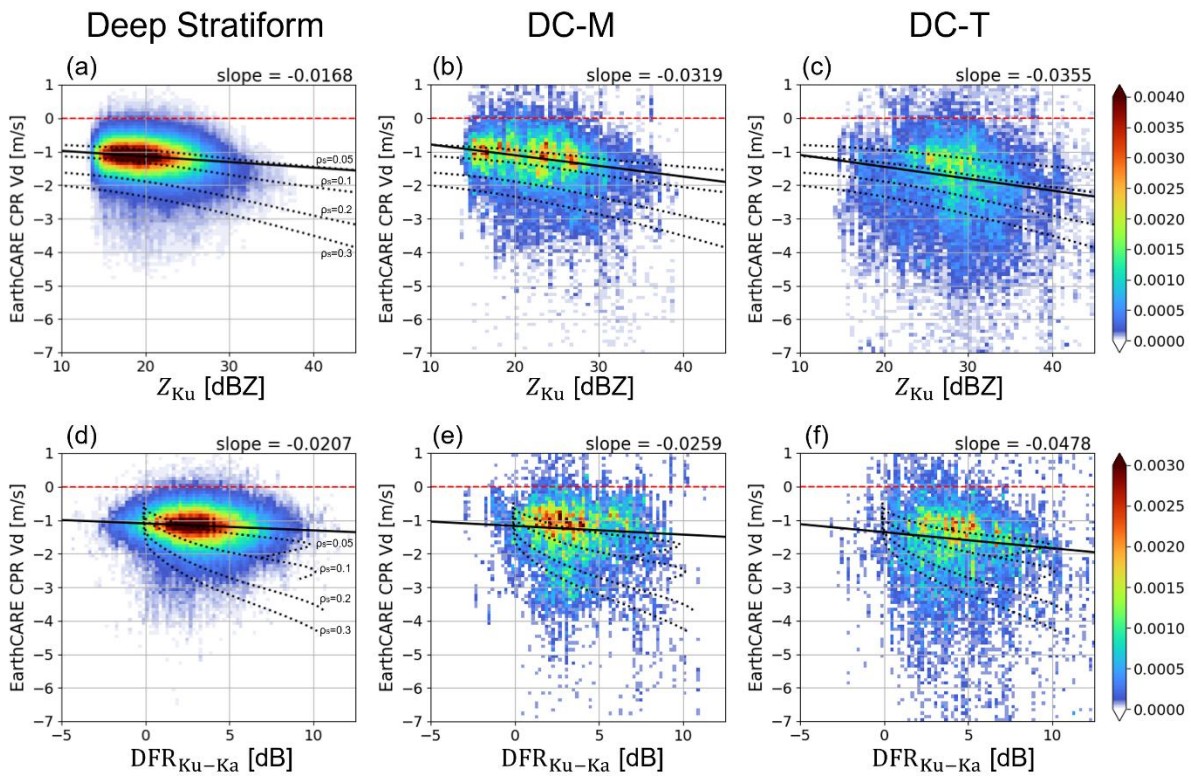

**Figure 9: (a–c) Joint histograms of $Z_{Ku}$ and $V_d$ and (d–f) joint histograms of Ku-Ka band dual-frequency reflectivity ratio (DFR) and $V_d$ for temperature range from –10°C to 0°C for (a, d) deep stratiform, (b, e) DC-M, and (c, f) DC-T precipitation types. Each histogram is normalized by the total number of samples of each precipitation type. The solid black lines represent regression lines fitted using the least squares method, with its corresponding slope indicated in the upper right corner outside each panel. The dotted black lines are same as those in Fig. 5, except that the x-axis is replaced with DFR in (d–f).**

**Table 1: Correlation coefficients and sample number of the joint histograms of $Z_{Ku}-V_d$ and DFR$-V_d$ for each precipitation type shown in Fig. 9.**

| Variable used in x-axis | Indicator | Stratiform | DC-M | DC-T |
|---|---|---|---|---|
| $Z_{Ku}$ | Correlation coefficient | −0. 1807 | −0. 2240 | −0. 1146 |
| | Sample number | 776223 | 23936 | 29636 |
| DFR$_{Ku-Ka}$ | Correlation coefficient | −0.0447 | −0.0693 | −0.1266 |
| | Sample number | 272624 | 12837 | 13093 |

### 3.4.2 Rain layers

We focus here on the characteristics of $V_d$ and $V_{air}$ inside the rain layers. Figures 10a and 10b present joint histograms of $Z_{Ku}$ and $V_d$ for deep stratiform and deep convective types, respectively, using data only from regions where temperature exceeds 4°C to exclude echoes from melting particles. In deep stratiform precipitation (Fig. 10a), consistent with the relationship seen in Fig. 5h, larger $Z_{Ku}$ values correspond to greater downward $V_d$, indicating that terminal velocities increase with larger mean drop sizes. The dashed line in the figure represents the theoretical $Z_{Ku}-V_{tz}$ relationship obtained from Eq. (4) for typical stratiform clouds, and the observed data generally aligns with this curve. Given that drop size distributions naturally vary from case to case, Fig. 10c shows $V_{tz}$ estimated from the $D_m$ values retrieved from DPR, based on Eq. (4). Subtracting these $V_{tz}$ estimates from the observed $V_d$ yields vertical air velocity, shown in Fig. 10e. For stratiform precipitation, the retrieved vertical velocities cluster around 0 m s$^{-1}$, which is consistent with the canonical understanding that vertical motions in stratiform rain are generally weak and near neutral (Fig. 10e).

Compared to deep stratiform precipitation, the $Z_{Ku}-V_d$ histogram for deep convective precipitation exhibits substantially greater variability and generally higher reflectivity values (Fig. 10b). As the joint histograms for DC-M and DC-T showed no significant distinction in their overall structure (not shown), both categories were combined into a single convective group. In the joint histogram of $Z_{Ku}$ and $V_{tz}$ for the deep convective type (Fig. 10d), the dominant particles exhibit larger $D_m$ compared to stratiform type, leading to faster falling speed of $V_{tz}$, with some cases reaching as high as –7 m s$^{-1}$.

The corresponding $V_{air}$ (Fig. 10f) reveal that, as $Z_{Ku}$ increases, the mean vertical velocity tends to shift toward stronger updrafts, accompanied by increased variance. These results indicate that stronger precipitation, associated with higher $Z_{Ku}$, often occurs in environments with more intense updrafts and turbulence. Such upward motions may sustain hydrometeors in the growth region longer, thereby enhancing collision–coalescence processes. This interpretation is consistent with the $Z_{Ku}$ profiles shown in Fig. 8, where reflectivity continues to increase toward lower altitudes in convective cases, suggesting ongoing raindrop growth supported by strong upward air motions.

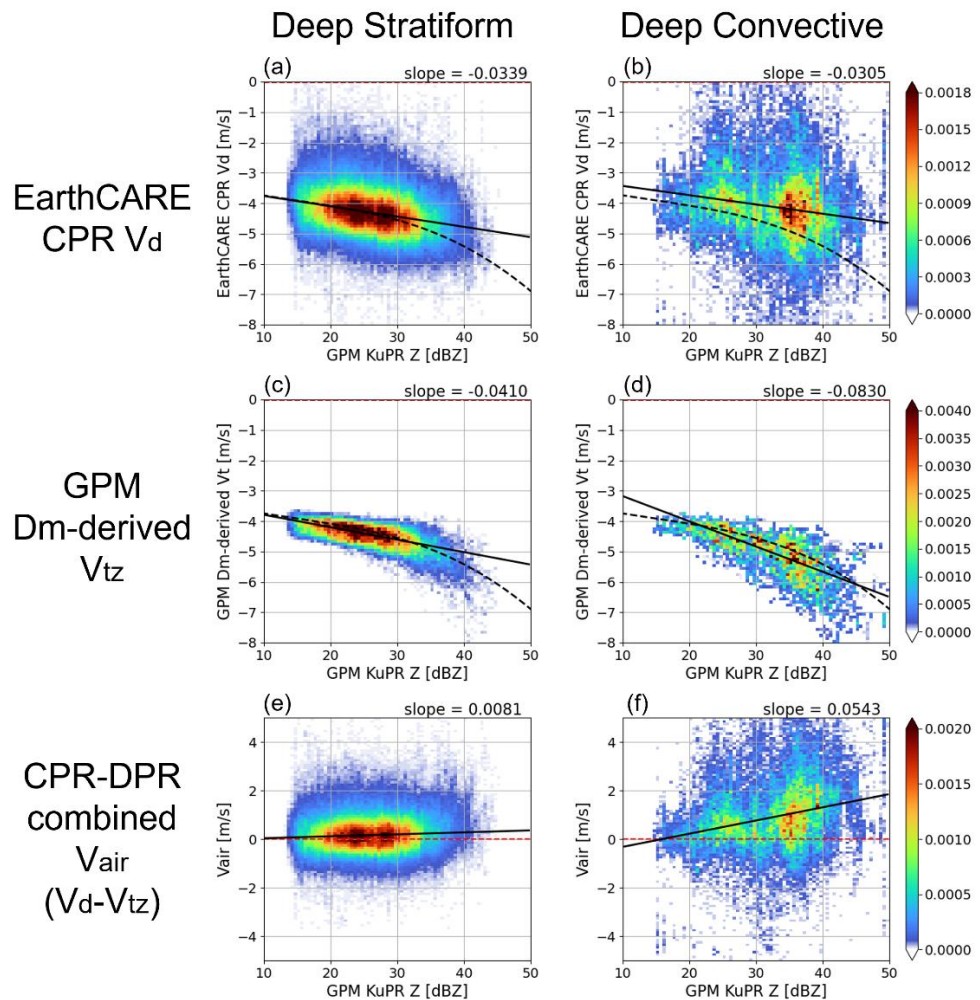

**Figure 10:** Joint histograms for the temperature above 4°C as a function of $Z_{Ku}$. The vertical axes represent (a, b) $V_d$ measured by CPR, (c, d) $V_{tz}$ and (e, f) $V_{air}$ calculated using the 2A.DPR product. (a, c, e) deep stratiform and (b, d, f) deep convective types. Each histogram is normalized by the total number of samples of each precipitation type. The solid black lines represent regression lines fitted using the least squares method, with its corresponding slope indicated in the upper right corner outside each panel. The dashed black lines in (a–d) are same as those in Fig. 5.

While a correlation between the DFR and $V_d$ was observed in the ice phase (Fig. 9 and Table 1), in typical rain layers the relationship between raindrop size and DFR becomes ambiguous due to the scattering nature of rain drops (Meneghini et al. 2022). Although using other frequency combinations, such as W- and Ka-band, could improve the correlation between DFR and particle size, interpretation becomes more difficult because of the strong attenuation and multiple-scattering effects at W-band, as shown in Fig. 5d. Therefore, the application of DFR in rain remains challenging and is left for future work.

**3.5 Comparison of vertical motions with single-sensor observations by the CPR**

CPR_CLP, one of JAXA's EarthCARE Level-2a standard products, provides its own estimates of $V_{air}$, which are derived solely from single-sensor observations by the CPR (Sato et al. 2025). Both the method implemented in CPR_CLP and the CPR-DPR combined approach presented in this study are based on the same fundamental concept of calculating $V_{air}$ by subtracting the $V_{tz}$ from the $V_d$. However, the two methods differ in how $V_{tz}$ is determined: the CPR_CLP estimates it from a particle size distribution (PSD) inferred using only the CPR-measured $Z_w$ and $V_d$, whereas the present method uses the PSD derived from the 2A.DPR algorithm. Because of this difference in the underlying PSDs and scattering database, discrepancies between the

two $V_{air}$ estimates are expected. Therefore, in this section, we compare the two $V_{air}$ estimates to assess the consistency and reliability of the retrieved $V_{air}$, as well as the implicit assumptions regarding $V_{tz}$ and PSD within each algorithm. Since the CPR_CLP product does not provide the quantity corresponding to $V_{tz}$, the comparison focuses solely on $V_{air}$.

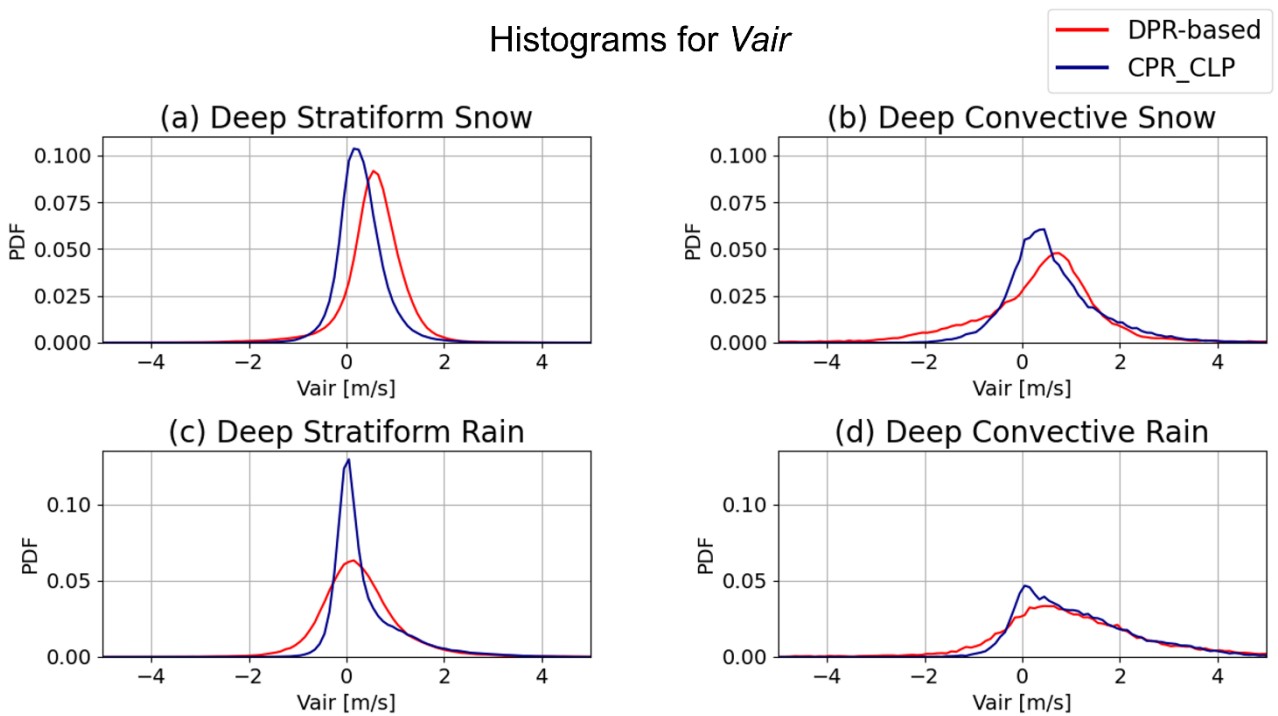

**Figure 11: Histograms $V_{air}$ calculated using the DPR -based method proposed in this study (red), and that using CPR_CLP (navy). (a, b) For the temperature range from –10°C to 0°C (snow) and (c, d) for the temperature above 4°C (rain). (a, c) deep stratiform and (b, d) deep convective types. Each histogram is normalized by the total number of samples for each precipitation type.**

**Table 2: The mean and standard deviation of each $V_{air}$ histogram shown in Fig. 11.**

| Indicator | $V_{air}$ retrieval method | Snow (–10°C < $T$ < 0°C) | Rain ($T$ > 4°C) |
|---|---|---|---|

| | | a) Stratiform | b) Convective | c) Stratiform | d) Convective |
|---|---|---|---|---|---|
| Mean | DPR-based | 0.561 | 0.356 | 0.279 | 1.112 |
| | CPR_CLP | 0.302 | 0.621 | 0.375 | 1.126 |
| Standard deviation | DPR-based | 0.638 | 1.366 | 0.874 | 1.631 |
| | CPR_CLP | 0.487 | 0.945 | 0.738 | 1.198 |

Figure 11 shows histograms of $V_{air}$ for deep stratiform and deep convective precipitation, separately for the temperature ranges corresponding to snow ($-10\ °C < T < 0\ °C$) and rain ($T > 4\ °C$). Table 2 summarizes the mean and standard deviation of the $V_{air}$ histograms shown in Fig. 11 for each retrieval method. Overall, the histograms exhibit smaller standard deviation in stratiform cases (0.5–0.9 m s⁻¹) and larger standard deviation in convective cases (0.9–1.6 m s⁻¹), likely associated with turbulent motions.

For the snow layers (Figs. 11a and 11b), weak upward motions appear on average in both types (0.3–0.6 m s⁻¹), likely associated with latent heat release during ice particle growth. On the other hand, some discrepancies exist between the two $V_{air}$ estimates, probably due to the radar frequency and assumptions about ice particle properties. The CPR_CLP algorithm accounts for scattering from ice particles with various shapes and orientations, whereas the DPR algorithm assumes simple spherical particles with a fixed bulk density of 0.10–0.13 g cm⁻³. The peak of the $V_{air}$ distribution from the DPR-based method is about 0.3 m s⁻¹ higher than that from CPR_CLP, suggesting a positive bias. This difference may indicate that the DPR-based PSD retrieval does not fully capture the contribution from smaller particles with slow $V_{tz}$ compared with the W-band CPR-based retrieval. On the other hand, for convective precipitation, the DPR-derived $V_{air}$ shows a larger proportion of downward motions ($< -1$ m s⁻¹) than the CPR_CLP $V_{air}$. This is likely because convective cases include more dense particles, such as graupel or hail, with densities exceeding 0.3 g cm⁻³, which are not represented in the current 2A.DPR algorithm.

For the rain layers (Figs. 11c and 11d), the DPR-based method and CPR_CLP show more similar $V_{air}$ histograms than in the snow cases. For stratiform precipitation, the mean $V_{air}$ is close to 0 m s⁻¹, whereas for convective precipitation, it shows stronger upward motion of about 1 m s⁻¹ with a larger variance. The two histograms agree well for strong upward motion ($V_{air} > 1$ m s⁻¹); however, for $V_{air} < 1$ m s⁻¹, the CPR_CLP $V_{air}$ shows a distinct peak around 0 m s⁻¹, while the DPR-based $V_{air}$ exhibits a smoother distribution. Despite such several differences, both methods yield comparable $V_{air}$ histograms overall.

While further validation is still needed, the approach presented in this study provides useful reference information for evaluating the validity of vertical air motion retrieval, which can otherwise only be obtained by specific ground-based vertically pointing radars where direct observational data are extremely limited.

## 4 Summary and discussions

With the advent of Doppler velocity measurements from the EarthCARE/CPR, it has become possible to observe the vertical motions of hydrometeors inside cloud and precipitation globally. While W-band radar observations by CPR are well-suited for detecting cloud particles and upper-level ice hydrometeors, Ku- and Ka-band radar observations by GPM/DPR are more effective under conditions involving rain or moderate-to-heavy ice precipitation, where attenuation and multiple scattering hinder reliable reflectivity measurements by CPR. In this study, we constructed a coincidence dataset of the EarthCARE/CPR and the GPM/DPR, and by integrating their complementary information, we investigated the characteristics of hydrometeor fall speeds and vertical air motion within precipitation systems.

We first performed two case studies of stratiform and convective precipitation events. An examination of the vertical profiles of radar reflectivity showed that while the DPR detected large raindrops and snow particles in advanced stages of growth, the CPR captured finer-scale cloud structures at higher altitudes. In the stratiform case, $V_d$ observations by CPR indicated slower downward motion above the bright band detected by the DPR and stronger downward motion below it. In contrast, the convective case exhibited considerable variability in $V_d$, with no clear transition near the melting level. In particular, in regions where $Z_{Ku}$ exceeded 30 dBZ, large downward $V_d$ values greater than 2 m s$^{-1}$ were observed above the freezing level, suggesting the presence of densely rimed particles such as graupel or hail.

To examine the relationship between $Z$ and $V_d$, we conducted a statistical analysis using nearly one year of data from August 2024 to June 2025. Above the freezing level, $Z$ in all frequency bands (W, Ku, and Ka) and $V_d$ generally increase with decreasing altitude, indicating that ice particles grow and become larger. In the upper troposphere above −10°C height, $V_d$ tends to increase with increasing $Z_W$, suggesting that higher reflectivity corresponds to faster-falling particles. However, in the lower layers below −10°C where snow has grown or melted into rain, the relationship between $Z_W$ and $V_d$ becomes less clear due to attenuation and Mie scattering effects in CPR observations. In contrast, although $Z_{Ku}$ lacks the sensitivity to detect precipitation above −10°C height, it exhibits a clear negative correlation with $V_d$ below −10°C height, effectively capturing the characteristics of hydrometeors in this layer. Furthermore, the slope of the regression line between $Z_{Ku}$ and $V_d$ is steeper in the liquid phase than in the solid phase, consistent with the fact that raindrop fall speeds are more sensitive to particle size than those of snow. These results suggest that $V_d$ can complement W-band reflectivity for estimating precipitation intensity in melting and rain layers where attenuation reduces the reliability of reflectivity alone.

Based on the classification of precipitation types using CTH and PTH, we identified four types: Deep Stratiform DC-M, DC-T, and Shallow. Distinct differences in the vertical profiles of $Z$ and $V_d$ were observed among these types. Above the freezing level, DC-T showed larger downward Doppler velocities and higher Ku-band reflectivity than deep stratiform and DC-M, supporting the presence of densely rimed particles such as graupel and hail. Convective cases (DC-M and DC-T) showed greater variability in both $Z$ and $V_d$ profiles, suggesting stronger turbulence compared to Deep stratiform cases.

In the rain layer, the $V_d$ of deep stratiform closely matched the $V_{tz}$ estimated from DPR, indicating that $V_{air}$ is near zero, which is consistent with the general characteristics of stratiform precipitation. In contrast, Deep convective precipitation showed

larger variability in $V_d$ and a tendency for $V_d$ to be larger than the estimated $V_{tz}$, particularly in regions with higher Ku-band reflectivity. This suggests that strong updrafts are present and is consistent with the interpretation that collision and coalescence processes are active, leading to an increase in $Z_{Ku}$ toward lower altitudes.

In addition, we compared the $V_{air}$ estimated using the DPR-derived PSD with that estimated solely from CPR measurements in the CPR_CLP product. The two estimates show consistent characteristics, exhibiting smaller variability with values concentrated around 0 m s$^{-1}$ for stratiform cases and larger variability for convective cases, likely associated with turbulent motions. On the other hand, a systematic bias is found between the two in the snow layer, which can be attributed to differences in the microphysical assumptions and observational characteristics due to radar frequency. Such information is expected to be valuable for improving both algorithms and for providing reference data to validate vertical velocity retrievals, which are otherwise extremely limited in direct observations.

The first-ever trial combining $V_d$ from CPR with three-frequency $Z$ from CPR and DPR provides vertical velocity information consistent with established knowledge of stratiform and convective precipitation systems. This highlights the potential of EarthCARE's spaceborne Doppler measurements to support a fundamental shift from conventional $Z$-based spatial pattern classification toward process-oriented classification, and to enhance our understanding of precipitation dynamics and microphysical evolution on a global scale. The results offer valuable insights for global precipitation characterization and support future algorithm development involving $V_d$, such as those under the EarthCARE CPR and the planned Ku-band Doppler precipitation radar under the Precipitation Measuring Mission (Nakamura and Furukawa, 2023).

Further research could include estimating ice particles using three-frequencies (Ku, Ka, and W-bands). Previous studies have demonstrated the potential of such approaches using the CloudSat/CPR and the DPR (Leinonen et al., 2015; Yin et al. 2017; Turk et al. 2021). CloudSat was in Daylight-Only Operations while the GPM Core Observatory was in operation. The EarthCARE-GPM coincidence dataset allowed for day and night analysis, which was impossible in the CloudSat-GPM coincidence dataset. In addition, studies using GMI in particular is expected to be an extension of this research in the future.

The Doppler velocity capability of the CPR can lead to improvements of the snowfall estimation using the GMI. While this work did not consider latitudinal variations, future data accumulation is expected to allow more refined classification of precipitation type according to meteorological systems, such as deep tropical cloud systems and snow storms in higher latitudes. Furthermore, synergy between the GPM sensors and the other EarthCARE sensors developed by European Space Agency (ESA) can be expected in future works.

*Data availability*

EarthCARE-GPM coincidence dataset is publicly available on the JAXA website under the Data DOI, https://doi.org/10.57746/EO.01ka7xakvwj6pcthxkvgt0vr0y. All EarthCARE products (JAXA, 2024a; JAXA, 2024b) and GPM products (JAXA, 2014) used in this study can be downloaded from the JAXA G-Portal.

*Author contributions*

SA performed the data analysis and drafted the paper. TK provided feedback on the analysis methods as well as on the manuscript draft. FJT provided the code for searching coincidence events and helped developing EarthCARE–GPM coincidence dataset.

*Competing interests*

The contact author has declared that none of the authors has any competing interests.

*Acknowledgements*

This work was supported by the 4th Research Announcement on the Earth Observations of the Japan Aerospace Exploration Agency (JAXA) (ER4GPF012). The work by FJT was carried out at the Jet Propulsion Laboratory, California Institute of Technology, under a contract with the National Aeronautics and Space Administration (80NM0018D0004). We would like to thank Dr. Kaya Kanemaru (NICT) and Mr. Taisei Tsuji (JAXA/EORC) for providing ideas regarding the estimation of vertical air velocity from CPR Doppler velocity using mean drop diameter derived from DPR.

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
