# Peer review of "Exploring vertical motions in convective and stratiform precipitation using spaceborne radar observations: Insights from EarthCARE and GPM coincidence dataset"

_EGUsphere, 2025_

## Author Comment (AC1)

**Response to community comment**

CC1: 'Comment on egusphere-2025-3596', Ousmane O. Sy

Thank you very much for carefully reading the manuscript and for providing such thoughtful and insightful comments. Based on your feedback, I have revised the manuscript to improve its clarity and to incorporate the changes in the revised submission. My point-by-point responses are provided below in red. Please note that the line numbers indicated in our responses refer to a revised version of the manuscript.

This article presents very interesting results based on a triple-frequency dataset complemented by Doppler measurements of the ESA/JAXA EarthCARE mission. It shows the high potential of coincident multi-frequency remote sensing observations with reflectivity, Doppler and passive microwave measurements. Such superdatabase can definitely help studies of dynamic atmospheric processes. The thermal analysis of the Doppler measurements is also eye-opening.

Thank you for recognizing the importance of our study. We plan to make the dataset, including passive microwave observations by GMI radiometer, publicly available from the JAXA server in the near future, and we hope that these data will contribute to a wide range of future analyses in atmospheric science.

My minor comments are detailed below.

**Comments:**

1. Equation 1: Given the convention to represent updrafts as positive velocities, I think the equation should be

In this study, positive values of Doppler velocity  $(V_d)$  are defined as upward motion. Likewise, we define the vertical upward direction as positive for both  $V_t$  and  $V_{\rm air}$ , so the fall velocity  $V_t$  is always negative, as shown in Eq. (5) and Eq. (7). Therefore, we believe that the formulation in Eq. (1) is correct as it stands. In the CPR L2a cloud product (CPR\_CLP), terminal fall velocity is also defined as negative, and to ensure consistency with that product, we would like to maintain this definition. We have added the following explanation regarding the negative sign of  $V_t$ :

Line 194

In this paper, positive values of  $V_d$  are defined as upward direction. Therefore, positive  $V_{air}$  means upward air motion and  $V_t$  is always negative.

2. Equation 2: the denominator is directly proportional to the reflectivity factor, but there is the factor (¥lambda^4/¥pi^5|K\_W|^2) missing, unless it is implicit in the definition of ¥sigma\_b?

As the reviewer correctly pointed out, a factor of  $\lambda^4/\pi^5|K_w|^2$  must be applied when calculating the reflectivity

factor. We have added the explanation, as shown below:

**Line 198**

The denominator of Eq. (2), when multiplied by  $\lambda^4/\pi^5|K_w|^2$ , where  $\lambda$  is the radar wavelength and  $K_w$  is the normalizing dielectric factor, corresponds to the radar reflectivity factor (Z).

3. Equation 1 represents all the error terms as additive (¥epsilon), which is a simplification. In fact there is also a multiplicative factor (which includes the randomness of the signal) that can be mitigated only by adaptive filtering, or along-track integration. Could you please include this caveat?

Thank you for this important comment regarding Doppler errors. As you pointed out, the term  $\varepsilon$  includes not only systematic biases but also random uncertainties arising from measurement noise. We have added the following caveat:

Line 203-205

In Eq. (2),  $\varepsilon$  is expressed as an additive term, but it includes not only systematic biases, but also random uncertainties mentioned above that can only be mitigated by adaptive filtering or along-track integration.

4. Line 211: It is true that the Doppler velocity comes from the phase change of the lag-1 correlation function, which in turns is not affected by attenuation. However, the magnitude of this correlation function (module of a complex number) is important to have reliable Doppler. Otherwise, the Doppler is all salt-and-pepper.

Thank you again for this important comment. We agree that, under conditions of strong attenuation or multiple scattering, the correlation may decrease to the point where Doppler information can no longer be retrieved. We believe that such cases are largely excluded by applying the multiple scattering filtering criteria of Battaglia et al. (2011), and therefore the impact on the statistics shown in Section 3 is expected to be small. However, we acknowledge that the original wording could cause misunderstanding, so we have revised it as follows:

Line 222-229

In contrast,  $V_d$  is retrieved from the pulse-to-pulse phase difference rather than from signal amplitude (Eisinger et al. 2024), and is therefore intrinsically less affected by attenuation of returned power (Doviak and Zrnic, 1993). In practice, pulse-to-pulse phase correlation is maintained under moderate attenuation, allowing the velocity retrievals to remain stable (Tian et al., 2007; Kollias et al., 2014). The main limitation arises when severe attenuation and multiple scattering associated with heavy rain or ice precipitation substantially degrades the pulse-to-pulse phase correlation, leading to large errors (Matrosov, 2008; Battaglia et al., 2011). In this study, cases containing rain, wet snow, and graupel were retained, while severe attenuation were excluded by applying screening following Battaglia et al. (2011), leading to preserve physically meaningful velocity information in these hydrometeor regimes.

5. Line 225: Does the use of Mie mean that it does not account for the shape/density variation of the frozen hydrometeors and their non-sphericity? Is that considered negligible microphysical variation?

For frozen hydrometeors, we assume homogeneous spherical particles with a fixed density. As you pointed out, it is indeed important to consider the shape and density variations of ice particles when evaluating

backscattering and terminal fall velocities. Significant differences are expected among snow, graupel, and hail. However, CPR observations, which are limited to the nadir direction, inherently provide no information to constrain the spherical assumption used here. This contrasts with ground-based dual-polarization radars, which observe the hydrometeor from the side, where particle asymmetry is more evident and can provide additional information. In this study, we therefore focus on rainfall  $V_t$ , where the spherical assumption is reasonable, and emphasize that the ice phase requires further investigation. Addressing ice particle shape and density variations constitutes an advanced research topic on its own, and we treat it as future work. We hope that this issue will be addressed in our own and other researchers' future studies. We revised the description to this point as following:

**Line 237-257**

For rain layers,  $v_t$  was computed using the empirical relationship proposed in Atlas and Ulbrich (1977), with a correction factor for air density, as given:

$$v_t(D) = -3.78D^{0.67} \cdot c(\rho), \tag{5}$$

$$c(\rho) = \sqrt{\frac{\rho_0}{\rho}} = \sqrt{\frac{\rho_0 RT}{p}}.$$
 (6)

Here, the unit of D is millimeters,  $\rho$  denotes the ambient air density,  $\rho_0$  is the standard air density (set to 1.225 kg m-3), R is the specific gas constant for dry air (287 J kg-1 K-1), and p and T represent pressure and temperature obtained from auxiliary data. The backscattering cross-section  $\sigma_b$  was derived from Mie scattering calculations for spherical raindrops at W-band frequency.

For snow,  $\sigma_b$  and  $v_t$  was calculated in the same manner as in the 2A.DPR algorithm, assuming homogeneous spherical particles with a density of 0.10–0.13 g cm-3 and a melted-equivalent diameter following the particle size distribution given by Eq. (3). The terminal fall velocity of snow was calculated following Magono and Nakamura (1965) as follows:

$$v_t(D_s) = -8.8(0.1D_s\rho_s)^{0.5} \cdot c(\rho), \tag{7}$$

where  $D_s$  is the unmelted snow particle diameter in mm, and  $\rho_s$  is the density of snow particles in g cm-3. On the other hand, ice particles can take various shapes, sizes, and densities, such as those of snow, graupel, and hail. Because  $\sigma_b$  and  $v_t$  vary depending on these parameters, the assumptions made for snow in this study are often not valid. Although it would be ideal to account for more realistic and complex scattering and fall characteristics of ice particles (Kuo et al. 2016; Ori et al. 2021), considering such diversity is challenging because the CPR observes only in the nadir direction and therefore cannot provide information on particle asymmetry. This contrasts with ground-based dual-polarization radars, which observe the hydrometeor from the side, where particle asymmetry is more evident and can provide additional information. In addition, such information on particle diversity cannot be inferred from the current version of the 2A.DPR algorithm and is therefore left for future work.

**6. Figs 2 and 3 are really great:**

- 1. it's great to see the various Z fields from DPR and ECPR, plus the Doppler!
- 2. The long along-track integration (10 km?) really helps to clean up aliasing in the Doppler and some of the NUBF in Fig.2;
- 3. However, doesn't it also lead to an "over-smoothing" of the Doppler for the convective scene in Fig.3?

Thank you for this valuable comment. First, we would like to clarify that the data included in the coincidence dataset are identical to those in the original EarthCARE/CPR L1b dataset, which provides 500 m horizontally

integrated values. The 10 km horizontal integration introduced in this study was an arbitrary length chosen for analytical purposes.

The horizontal 10 km integration is highly effective in reducing the contribution of random noise induced by decorrelation. However, as you pointed out, in convective precipitation this integration may lead to over-smoothing, potentially mixing the signatures of strong echoes within convective cores with weaker echoes at cloud edges.

In response to comments from another reviewer, we have shortened the integration length to 5 km. As a result, the Doppler velocity profiles in Fig. 3d now show reduced horizontal stripe patterns that were present in the original version, while still effectively suppressing random noise and the NUBF effect. This modification alleviated the over-smoothing issue.

Original Fig. 3d. Vertical cross section of CPR Doppler velocity with 10-km horizontal integration.

Updated Fig. 3d. Vertical cross section of CPR Doppler velocity with 5-km horizontal integration.

It should be noted, however, that since the integration is performed as a reflectivity-weighted average rather than a simple moving average, the convective core features tend to be emphasized. This may lead to artificially enhanced downward velocities near cloud edges where echoes are weak. In the statistical analyses presented in Sections 3.2 and 3.3, such edge regions, where the coincidence with DPR observations could not be ensured after averaging, were excluded from the analysis, as indicated by the black-plotted areas in Fig. 3g. Therefore, we believe that the impact of over-smoothing has been effectively mitigated in our results. We have also added the following description:

**Line 285-292**

The horizontal 5 km integration applied to the  $V_d$  field in Fig. 2d is highly effective in reducing the contribution of random noise induced by decorrelation ( $\varepsilon_{random}$ ). However, in the convective case shown in Fig. 3d, this integration may result in over-smoothing, mixing the signatures of strong echoes within convective cores with weaker echoes at cloud edges. Because the integration is performed as a reflectivity-weighted average rather than a simple moving average, the features of the convective core are emphasized, which may in turn lead to artificially enhanced downward velocities near cloud edges where echoes are weak. In the statistical analyses presented later in this paper, such edge

regions, where coincidence with DPR observations could not be ensured after averaging, were excluded from the analysis, as indicated by the black-plotted areas in Figs. 2g and 3g.

7. L.305: Isn't it "CFED of Vd shown in Fig. 4b illustrates an increase in downward velocity with decreasing increasing temperature between -20°C and 0°C,"?

Thank you for pointing this out. The term "decreasing" was incorrect and should be replaced with "increasing."

- 8. The CFEDS of the W band are also great!
  - 1. Fig.4a shows the stagnation of Z around [-10,0]^oC (due to competing effects of increasing unattenuated Z due to growth of particle, and, increasing attenuation) and
  - 2. the slightly increasing Vd in that range shows that there is indeed a growth of particles,

That is exactly the point we have also been focusing on. We believe that the advantage of observing not only Z but also  $V_d$  becomes evident in that temperature range. Although this study does not go into that level of detail, we hope that future work involving scattering calculations for snow and ice particles will allow us to relate these observations to temperature-dependent variations in particle shape and density.

- 1. The scatter-plots in Figs 5, 9 and 10 show the various datasets together.
  - 1. Would it help to show the concentration in log(counts) instead of counts? It may be that there aren't enough points to consider a log scale...

We also plotted the color shades in Figs. 5, 9, and 10 on a log scale (Figs. C1, C2, and C3, respectively). The visibility of the figures did not change much compared to the originals. Since plotting in log scale makes it less intuitive to grasp the count values, we prefer to keep the original figures.

Figure C1: Same as Fig. 5, but with log-scale color shade.

Figure C2: Same as previous Fig. 9, but with log-scale color shade.

Figure C3: Same as Fig. 10, but with log-scale color shade.

1. L.429 and 501: rimmed or rimed? (please correct various instances in the article).

Thank you. We have revised the manuscript accordingly.

- 2. I was wondering if it would be worth showing plots/statistics of the DWR (ZKu-ZKa) as well?
  - 1. it is directly related to Dm, which plays a central role in the PSD used to estimate Vt in Eqs 3 and 4 and
  - 2. it is a clear indicator when the attenuation of Ka is excessive.

Thank you very much for this important comment. The dual-frequency ratio (DFR or DWR) is indeed theoretically more directly related to the particle size distribution than the reflectivity alone. In response, we have added a diagram in Fig. 9 for the ice phase, using DFR ( $Z_{Ku}/Z_{Ka}$ ) on the x-axis, and included the following description:

Line 507-520

In some previous studies, the dual-frequency reflectivity ratio (DFR) has been used to characterize ice-phase precipitation (Leinonen et al., 2015; Yin et al. 2017; Akiyama et al. 2025). Compared with using single-frequency Z, DFR cancels the uncertainty associated with the number concentration  $N_W$ , thereby is more directly related to the particle size distribution and attenuation. Figures 9d–9f show joint histograms with the Ku–Ka band DFR ( $Z_{Ku}/Z_{Ka}$ ) from the DPR plotted on the x-axis. Here, we used Z which was not corrected for attenuation. As in the discussion using  $Z_{Ku}$ , the convective type tends to show larger DFR values and faster downward  $V_d$  compared to the stratiform type, with a steeper regression slope in the DFR- $V_d$  relationship. This suggests the dominance of larger particles with higher density. However, correlation coefficients for each case (Table 1) show that  $Z_{Ku}$  correlates more strongly with  $V_d$  than DFR does. This simple analysis therefore does not demonstrate a clear advantage of using DFR. The spread of  $V_d$  with respect to DFR may reflect the variations in microphysical characteristics such as particle shape and density, as well as atmospheric turbulence. Moreover, because the DFR was calculated using the KaPR HS observation swath, the number of samples is about half that of Figs. 9a–9c, which may have resulted in the lower correlation. As future work, once a larger multi-year dataset becomes available, scattering calculations that account for variations in ice particle shape and density will enable DFR to provide more detailed insights into cloud and precipitation microphysics.

On the other hand, for rain, DFR ( $Z_{Ku}/Z_{Ka}$ ) tends to take small negative values close to zero around Dm = 0.5–1.5 mm (Fig.1 in Meneghini et al. 2022), which is the range often used as the representative mean raindrop diameter. In this range, the relationship between DFR and particle size becomes ambiguous. While the correlation between DFR and particle size can be improved by using other frequency combinations (e.g., W-and Ka-band), in such cases the strong attenuation and multiple scattering at W-band would need to be carefully addressed. Therefore, the application of DFR in rain remains challenging and is left for future work. We have added the following explanation to the manuscript:

Line 561-566

While a correlation between the DFR and  $V_d$  was observed in the ice phase (Fig. 9 and Table 1), in typical rain layers the relationship between raindrop size and DFR becomes ambiguous due to the scattering nature of rain drops (Meneghini et al. 2022). Although using other frequency combinations, such as W- and Ka-band, could improve the correlation between DFR and particle size, interpretation becomes more difficult because of the strong attenuation and multiple-scattering effects at W-band, as shown in Fig. 5d. Therefore, the application of DFR in rain remains challenging and is left for future work.

Figure 1: (a–c) Joint histograms of  $Z_{Ku}$  and  $V_d$  and (d–f) joint histograms of Ku-Ka band dual-frequency reflectivity ratio (DFR) and  $V_d$  for temperature range from  $-10^{\circ}$ C to  $0^{\circ}$ C for (a, d) deep stratiform, (b, e) DC-M, and (c, f) DC-T precipitation types. Each histogram is normalized by the total number of samplings of each precipitation type. The solid black lines represent regression lines fitted using the least squares method, with its corresponding slope indicated in the upper right corner outside each panel. The dotted black lines are same as those in Fig. 5, except that the x-axis is replaced with DFR in (d–f).

Table 1: Correlation coefficients and sample number of the joint histograms of  $Z_{Ku}$ – $V_d$  and DFR– $V_d$  for each precipitation type shown in Fig. 9.

| Indicator          | V air retrieval method | Snow $(-10^{\circ}\text{C} < T < 0^{\circ}\text{C})$ |               | Rain $(T > 4$ °C) |               |
|--------------------|-----------------------------------|------------------------------------------------------|---------------|-------------------|---------------|
|                    |                                   | a) Stratiform                                        | b) Convective | c) Stratiform     | d) Convective |
| Mean               | DPR-based                         | 0.561                                                | 0.356         | 0.279             | 1.112         |
|                    | CPR_CLP                           | 0.302                                                | 0.621         | 0.375             | 1.126         |
| Standard deviation | DPR-based                         | 0.638                                                | 1.366         | 0.874             | 1.631         |
|                    | CPR_CLP                           | 0.487                                                | 0.945         | 0.738             | 1.198         |

1. In the interpretation of Fig.8, the larger standard deviation of the Doppler is attributed to turbulence. Couldn't it be caused also/instead by 1) shear or 2) the microphysical variability of Vt? Or are these implied in the term "turbulence"?

We agree that shear and microphysical variability are also important factors, and it is better to state this explicitly. We have therefore revised the text as follows:

**Line 490-492**

Furthermore, the larger standard deviation of  $V_d$  compared to that in stratiform precipitation indicates more active turbulent motion, including contributions from vertical air motion, wind shear and microphysical variability.

---

## Author Comment (AC2)

**Response to referee**

RC1: 'Comment on egusphere-2025-3596', Anonymous Referee #1

General Comments: To demonstrate the utility of the newly available EarthCARE data, the authors in this study investigate the EarthCARE CPR radar measured Doppler velocity in convective and stratiform precipitating clouds and interpret it in relation to precipitation growth processes. The uniqueness of this study is the combined use of both EarthCARE CPR and GPM DPR data, which allows the authors to explore the difference of particle growth processes in stratiform and convective clouds. The paper seems to serve 2 purposes: First, it demonstrates the quality and usefulness of the first-ever space-borne cloud radar Doppler velocity measurements, and second, using the Doppler velocity measurements, it confirms some of the understandings on microphysical processes in convective and stratiform clouds. The paper is well structured, and the messages are well presented. It will be a good contribution to this special collection of papers on EarthCARE. I suggest accepting after addressing some minor concerns.

The authors sincerely thank the reviewer for carefully reading the manuscript and for providing constructive feedback. We also appreciate the reviewer's recognition of the value of our study. We have carefully reviewed the manuscript in response to the reviewer's comments. My point-by-point responses are provided below in red. Please note that the line numbers indicated in our responses refer to the revised manuscript.

**Specific Comments:**

1. Doppler velocity Vd in the EarthCARE product. In the manuscript, the authors stated that the Doppler velocity in the EarthCARE product is derived from "the phase shift of the radar signal and is therefore less affected by attenuation". I understand that the retrieval of Vd is not the focus of this paper, but I do like to see a brief explanation on this topic in the Data and Methods section. Because many of the cases involved in this study are related to moderate to heavy rainfall, CPR should suffer significant attenuation particularly for the rain portion in the vertical profiles. Are there any studies on the impact on Vd retrieval accuracy by attenuation when using phase shift method?

Thank you for this important suggestion. We have added an explanation about the principle and previous studies showing that Doppler velocity is less affected by attenuation than radar reflectivity. It is also known that the reliability of Doppler velocity decreases under conditions of strong attenuation; such cases were excluded from the statistical analysis in this study. The original description could be misinterpreted as implying that Doppler velocity is completely unaffected by attenuation, so we have revised the text to clarify these points as follows.

**Line 222-229**

In contrast,  $V_d$  is retrieved from the pulse-to-pulse phase difference rather than from signal amplitude (Eisinger et al. 2024), and is therefore intrinsically less affected by attenuation of returned power (Doviak and Zrnic, 1993). In practice, pulse-to-pulse phase correlation is maintained under moderate attenuation, allowing the velocity retrievals to remain stable (Tian et al., 2007; Kollias et al., 2014). The main limitation arises when severe attenuation and multiple scattering

associated with heavy rain or ice precipitation substantially degrades the pulse-to-pulse phase correlation, leading to large errors (Matrosov, 2008; Battaglia et al., 2011). In this study, cases containing rain, wet snow, and graupel were retained, while severe attenuation were excluded by applying screening following Battaglia et al. (2011), leading to preserve physically meaningful velocity information in these hydrometeor regimes.

2. 10 km horizontal integration. Please explain the reason for 10 km integration for EarthCARE data. The DPR footprint size is about 5 km, then you have to average 2 DPR pixels to match 13 (=10./0.75) EarthCARE pixels? Is there a reason not using ~7 EarthCARE pixels to match 1 DPR pixel? Will the results be different if do so? In short, the decision to use 10 km seems to be somewhat arbitrary, may need couple sentence to justify.

In this study, a 10 km along-track integration was originally applied for each CPR grid point to mitigate the effects of footprint differences between CPR and DPR and to reduce errors contaminating the Doppler velocity. There is a trade-off in selecting the integration length: a longer integration reduces random errors and small-scale natural fluctuations in  $V_d$ , but if it is too long, the smoothing can become excessive and the observed features may no longer correspond spatially. To assess this, we reanalyzed all data using a 5 km integration. Although the variance slightly increased, the overall results remained essentially unchanged. Considering consistency with the DPR footprint as well, we decided to adopt the 5 km integration and replaced all original Figures 2–10 with the updated ones.

Based on the above, we have added the following explanation regarding the integration length to the manuscript. We also clarified that, although the native footprint diameter is about 750 m, the product provides data at 500 m intervals along the track.

**Line 128-134**

In this study, EarthCARE L1B CPR one-sensor products (JAXA, 2024a) from 1 August 2024 to 30 June 2025 was utilized, providing radar reflectivity factor and Doppler velocity products. Although the native footprint diameter is about 750 m, the product provides data at 500 m intervals along the track. The vertical resolution is 500 m with 100 m vertical grid spacing. In this study, 5 km along-track integration is applied for each CPR grid point using neighboring 10 grid points to mitigate the effects of footprint differences between CPR and DPR. This horizontal integration also helps reduce the errors that contaminate the Doppler velocity, as described in Section 2.2. The integrated data retains the original 500 m spacing, rather than being resampled at 5 km intervals.

3. The use of temperature as vertical coordinate. The use of temperature as vertical coordinate is an interesting way to investigate microphysical processes. However, there is a shortcoming when global data are mixed into one figure such as Figure 4. I suspect that most of data near 20C are from tropics or warm season mid-latitudes. In the meantime, data near 0C are from almost all the places. When we put all data into one figure, explaining the features in a way that particles are falling from aloft to lower part is somewhat misleading. I'd like the authors mention this shortcoming, and remind readers that future studies should separate data into groups with similar temperature range in the vertical.

Thank you for your careful comments and suggestions for future research. As you pointed out, the transition in the CFED diagram from 0°C to 20°C may not necessarily reflect vertical microphysical growth processes, but

could instead result from combining data from different geographical locations. To avoid potential misinterpretation, we have added the following text to clarify the aspects that this study does not address:

Line 356-360

It should be noted that although  $Z_{Ku}$  and  $Z_{Ka}$  increase with increasing temperature between 10 °C and 25 °C (Figs. 4c and 4d), this does not necessarily imply stronger precipitation lower in the column. Near the melting layer, the data include contributions from all latitudes, whereas the observations around 20 °C are dominated by low-latitude regions. Although the limited number of samples makes detailed discussion difficult at present, future work should involve analysis separated by weather systems and freezing level height to better investigate the vertical growth processes of hydrometeors.

4. Just a comment. It is great to see in Figure 5 that Vd in the cold range (-10C) is around -1 m/s and the derived Vt is matching well with measured Vd. This gives us great confidence that the Vd quality is high.

We thank the reviewer for providing this valuable comment from a new perspective. As you pointed out, the observed  $V_d$  of ice clouds in clod range is useful for comparison with the theoretically derived reflectivity-weighted terminal fall velocity ( $V_t$ ). Indeed, as seen in Fig. 5e of the previous version of the manuscript, the measured  $V_d$  appears to agree reasonably well with the theoretical  $V_t$ . However, in this temperature range, it is more appropriate to discuss the relationship using histograms of Z at W-band from CPR observations. Therefore, we have added theoretical  $Z_W$ – $V_t$  relationship curves to Figs. 5a–d.

Because the  $V_t$  of ice particles has been reported for various densities and shapes, there is ongoing discussion regarding which theoretical values should be adopted. The line shown in the original Fig. 5 was calculated under a specific condition and was therefore somewhat arbitrary. Therefore, to avoid misinterpretation, we have plotted theoretical curves for different densities (0.05, 0.1, 0.2, and 0.3 g cm-3). We have also revised the corresponding text related to this point in the manuscript accordingly:

Line 361-370

To examine the relationship between Z and  $V_d$  in more detail across different frequencies, histograms were constructed for various temperature ranges, as shown in Fig. 5. The dashed and dotted black lines in Fig. 5 represent the theoretical Z–  $V_t$  relationship for rain and snow, respectively, calculated under the assumptions described in Section 2.2. Figures 5a–5d use  $Z_W$  as the horizontal axis, while Figs. 5e–5h use  $Z_{Ku}$ . For snow, lines corresponding to  $\rho_s$  of 0.05, 0.1, 0.2, and 0.3 g cm-3 are plotted. Such Z– $V_d$  relationships have long been investigated using ground-based radar observations and serves as a useful metric for inclusion in weather and climate models. In the upper troposphere (T <  $-10^{\circ}$ C; Fig. 5a),  $V_d$  tends to increase with increasing  $Z_W$ , indicating that larger reflectivity is associated with faster-falling particles. Assuming that the vertical air motion averages to zero over many samples, the mean  $V_d$  can be interpreted as representative of the  $V_t$ . The  $Z_W$ – $V_d$  distribution in Fig. 5a follows the theoretical  $V_t$  curve for  $\rho_s$  = 0.05 g cm-3, showing that the downward fall speed increases with increasing  $Z_W$ , which provides insight into the growth of ice particles.

Figure 1: Joint histograms of radar reflectivity and Doppler velocity for four temperature ranges: (a, e) <  $-10^{\circ}$ C, (b, f) -10 to  $0^{\circ}$ C, (c, g) 0 to  $4^{\circ}$ C, and (d, h) >  $4^{\circ}$ C. Panels (a–d) use CPR radar reflectivity on the x-axis, while (e–h) use KuPR attenuation-corrected radar reflectivity. The red dashed line indicates 0 m/s Doppler velocity. The solid black lines in (e–h) represent regression lines fitted using the least squares method, with their corresponding slopes indicated in the upper right corner outside each panel. The dashed and dotted black lines represent the theoretical  $Z_{Ku} - V_t$  relationship for rain drops at 850 hPa and  $10^{\circ}$ C, and for snowflakes with a densities of 0.05, 0.1, 0.2 and 0.3 g cm-3 at 600 hPa and  $-10^{\circ}$ C calculated from eq. (4), where radar reflectivity in W-band and Ku-band is calculated assuming  $N_w = 10^3 \text{ mm}^{-1} \text{ m}^{-3}$ .

In addition, because the original manuscript did not explicitly describe how  $V_t$  for snow was calculated and did not discuss the limitations of the assumed particle properties, we have revised and added the following text to Section 2.2:

**Line 237-257**

For rain layers,  $v_t$  was computed using the empirical relationship proposed in Atlas and Ulbrich (1977), with a correction factor for air density, as given:

$$v_t(D) = -3.78D^{0.67} \cdot c(\rho_a),\tag{5}$$

$$c(\rho_a) = \sqrt{\frac{\rho_{a0}}{\rho_a}} = \sqrt{\frac{\rho_{a0}RT}{p}}.$$
 (6)

Here, the unit of D is millimeters,  $\rho_a$  denotes the ambient air density,  $\rho_{a0}$  is the standard air density (set to 1.225 kg m-3), R is the specific gas constant for dry air (287 J kg-1 K-1), and p and T represent pressure and temperature obtained from auxiliary data. The backscattering cross-section  $\sigma_b$  was derived from Mie scattering calculations for spherical raindrops at W-band frequency.

For snow,  $\sigma_b$  and  $v_t$  was calculated in the same manner as in the 2A.DPR algorithm, assuming homogeneous spherical particles with a density of 0.10–0.13 g cm-3 and a melted-equivalent diameter following the particle size distribution given by Eq. (3). The terminal fall velocity of snow was calculated following Magono and Nakamura (1965) as follows:

$$v_t(D_s) = -8.8(0.1D_s\rho_s)^{0.5} \cdot c(\rho_a), \tag{7}$$

where  $D_s$  is the unmelted snow particle diameter in mm, and  $\rho_s$  is the density of snow particles in g cm-3. On the

other hand, ice particles can take various shapes, sizes, and densities, such as those of snow, graupel, and hail. Because  $\sigma_b$  and  $v_t$  vary depending on these parameters, the assumptions made for snow in this study are often not valid. Although it would be ideal to account for more realistic and complex scattering and fall characteristics of ice particles (Kuo et al. 2016; Ori et al. 2021), considering such diversity is challenging because the CPR observes only in the nadir direction and therefore cannot provide information on particle asymmetry. This contrasts with ground-based dual-polarization radars, which observe the hydrometeor from the side, where particle asymmetry is more evident and can provide additional information. In addition, such information on particle diversity cannot be inferred from the current version of the 2A.DPR algorithm and is therefore left for future work.

**5. Misc.**

Line 55. "The Tropical ... (TRMM) was launched in 1997, and the TRMM carried ...". I think it is better to say: "The Tropical ... (TRMM) *satellite* was launched in 1997, and it carried ..."

We have revised the manuscript accordingly (Line 55).

Line 105. I don't see "CSATGPM" appearing in any place before this point. Please define it.

We added the following definition to Line 102:

CloudSat-GPM coincidence dataset (CSATGPM; Turk et al., 2021)

Line 137. The exclusion of 5 and 10 range bins are somewhat arbitrary. Are they about 0.5 and 1.0 km, respectively? Please add a couple of sentences to explain why excluding these many bins is enough.

We have added the answer to the reviwer's question to the manuscript as follows:

Line 142-146

Pre-launch studies suggested that the EarthCARE CPR would be less affected by surface clutter than the CloudSat CPR and that, over flat surfaces, clutter would not extend above 600 m (Roh et al., 2023). Observations are consistent with this expectation, although the altitude affected by clutter is higher over mountainous terrain. Taking these factors into account, we excluded data within five range bins (~500 m) above the ocean surface and ten range bins (~1000 m) above land to avoid potential contamination from ground clutter.

Line 164-165. Earlier in the text, it is mentioned that EarthCARE data is integrated to a 10 km "pixel". Here it sounds like the matching is between 1 EarthCARE original pixel (750 m size) with 1 DPR pixel (5 km size). Please clarify.

In this study, 5 km along-track integration is applied for each CPR grid point (with 500m interval) using neighboring 10 grid points. Consequently, the resulting data retains the original 500 m spacing, rather than being resampled at 10 km intervals. I have revised the manuscript to make this point clear.

Line 127-134

In this study, EarthCARE L1B CPR one-sensor products (JAXA, 2024) from 1 August 2024 to 30 June 2025 was utilized, providing radar reflectivity factor and Doppler velocity products. Although the native footprint diameter is about 750 m, the product provides data at 500 m intervals along the track. The vertical resolution is 500 m with 100 m vertical grid spacing. In this study, 5 km along-track integration is applied for each CPR grid point using neighboring 10 grid

points to mitigate the effects of footprint differences between CPR and DPR. This horizontal integration also helps reduce the errors that contaminate the Doppler velocity, as described in Section 2.2. The integrated data retains the original 500 m spacing, rather than being resampled at 5 km intervals.

Line 170-171

For each coincidence event, the nearest DPR footprint was matched to every CPR horizontal grid points with 500 m spacing.

Line 276-277. Do you have a rough number of profiles (in percent) that is detected by CPR but not DPR?

The following information has been added regarding the number of profiles:

Line 313-314

Only profiles where echoes are detected by both DPR and CPR are included. These profiles correspond to 5.1% of all profiles and 10.1% of the profiles in which echoes are detected by the CPR.

Figure 6. An interesting feature is that most stratiform precipitation tops (by DPR) are around -15C although their cloud tops (by CPR) are all over the place. Any explanations?

Thank you very much for your helpful comments. Indeed, the PTH of stratiform precipitation is concentrated within the temperature range of  $-10^{\circ}$ C to  $-20^{\circ}$ C, which is noteworthy. We have revised and added the following explanation and discussion to the manuscript:

Line 411-419

When the PTH is located above the  $0^{\circ}$ C level, it is indicative of a cold-type precipitation process, in which ice particles grow into relatively large snow aggregates or graupel through aggregation and riming. In stratiform precipitation, the PTH is mostly at temperatures lower than  $0^{\circ}$ C, indicating that most cases are associated with cold-type precipitation (Fig. 6a). In particular, a high frequency of occurrence is confined within the PTH range of  $-20^{\circ}$ C to  $-10^{\circ}$ C regardless of CTH. This temperature range corresponds to the layer where ice habits transition with temperature, as shown in the classical Nakaya diagram (Libbrecht 2005). It is also referred to as the dendritic growth layer, where cloud ice particles are thought to grow into snow through depositional growth, aggregation, and potentially secondary ice processes (von Terzi et al. 2022). Such temperature-dependent microphysical processes may explain why ice particles in typical stratiform clouds become large enough to be detected by the KuPR only when they reach below the  $-20^{\circ}$ C level.

Furthermore, unlike in stratiform cases, the PTH in convective precipitation can vary widely from about 20°C to -40°C. We have revised the following discussions to the manuscript to reflect this point:

Line 420-429

In contrast, convective precipitation shown in Fig. 6b exhibits a much wider range of PTH values, extending from 20°C down to below -40°C, with a sparse distribution in the CTH-PTH histogram. Focusing on deep clouds with CTH above the -20°C level, the PTH tends to lie close to the one-to-one line between PTH and CTH, indicating that the precipitation top height is nearly as high as the cloud top. This situation can be interpreted as the result of strong updrafts within the system that lifted large hydrometeors toward the cloud top, as discussed in Takahashi and Luo (2014). In addition, in convective cases, a pronounced peak in occurrence is found where the PTH is below the 0°C level. When the PTH is below the 0°C level, it suggests a warm-type precipitation process, where raindrops grow through collision and coalescence of liquid water droplets. Such warm-type shallow precipitation is likely associated mainly with shallow cumulus or congestus clouds, because the DPR has limited sensitivity to detect light precipitation (~1 mm h-1 or less; Hayden and Liu, 2018), and thus does not effectively capture shallow stratus or stratocumulus.

Line 447. "theoretical W-band terminal velocity" -> "theoretical terminal velocity". Terminal velocity should not be band-dependent.

Thank you for the comment. It is true that terminal velocity itself does not depend on radar frequency. However, the figure shows the reflectivity weighted terminal velocity ( $V_t$ ) obtained from Eq. (4), which is band-dependent. We have revised the manuscript to clarify this point by changing the expression "theoretical W-band terminal velocity" to "theoretical  $Z_W$ – $V_t$  relationship.", not only at the original Line 447 but also in other instances where similar wording was used throughout the manuscript.

---

## Author Comment (AC3)

**Response to referee**

RC2: 'Comment on egusphere-2025-3596', Anonymous Referee #2

This work analyses Ku-, Ka-, and W-band radar measurements of precipitating clouds using a one-year dataset of matchup EarthCARE and GPM observations. Following a quick demonstration of two case studies, statistical properties of radar reflectivity and Doppler velocity profiles are investigated for different particle phases (liquid/solid) and different precipitation types (shallow/convective/stratiform). A new method is devised to evaluate the vertical air motion in raining layers by subtracting the terminal velocity deduced by DPR-derived DSD from the EarthCARE Doppler velocity.

This is a well written paper presenting robust analysis results that align with theoretical expectations and physical intuition. The authors' effort to construct a matchup EarthCARE and GPM dataset should be applauded and will be welcomed by the cloud/precipitation science community. I suggest a few revisions that are mostly minor in nature with the possible exception of the first point. Otherwise I would recommend that the paper be published in AMT.

The authors sincerely thank the reviewer for carefully reading the manuscript and for providing constructive feedback. We also appreciate the reviewer's recognition of the value of our study. We have carefully reviewed the manuscript in response to the reviewer's comments. My point-by-point responses are provided below in red. Please note that the line numbers indicated in our responses refer to the revised manuscript.

Main comments -----

1. JAXA's EarthCARE CPR Level-2 cloud product (L2a CPR\_CLP) contains its own vertical air motion estimated from Doppler velocity. The CPR reflectivitiy (and hence DSD estimates crucial for V\_t and V\_air as well) is subject to heavy attenuation for intense rain as the authors pointed out (II. 210-211). That being said, a substantial number of CPR reflectivities would be still usable, being not entirely washed out by attenuation even beneath the 0-degree level as far as I can tell from Fig. 2c and 3c. This means that there would be plenty of simultaneous measurements available for both EarthCARE-provided V\_air and GPM Dm estimates.

I am curious how consistent the V\_t estimates are between the CPR standard product and the current method using DPR-derived DSD. You would find discrepancies because the CPR\_CLP relies on its own built-in DSD assumption which is not guaranteed to accord with DPR Dm. An additional plot or two comparing the CPR-only and DPR-based V\_t and V\_air estimates would tell us how reliable the CPR products are, offering useful information for EarthCARE algorithm developers and interested users.

We thank the reviewer for this insightful suggestion, which enhances the value of the estimated  $V_{\rm air}$  in our study. As you pointed out, the EarthCARE standard product, CPR\_CLP, estimates  $V_{\rm air}$  based on its own built-in DSD

assumptions and scattering database. Comparing these estimates with those derived from the DPR-based DSD provides useful information for both algorithm developers and users in evaluating the consistency and validity of each retrieval approach. Accordingly, we conducted a new analysis comparing  $V_{\rm air}$  retrievals from CPR\_CLP with our results. Figure 11 shows histograms of  $V_{\rm air}$  for deep stratiform and deep convective precipitation, separately for the temperature ranges corresponding to snow ( $-10~{\rm ^{\circ}C}$  < T < 0  ${\rm ^{\circ}C}$ ) and rain (T > 4  ${\rm ^{\circ}C}$ ). Along with, Table 2 shows the mean and standard deviation of each histograms shown in Fig. 11. Based on these results, we have added a discussion in new Section 3.5 to the revised manuscript as follows:

**Line 568-608**

**3.5 Comparison of vertical motions with single-sensor observations by the CPR**

CPR\_CLP, one of JAXA's EarthCARE Level-2a standard products, provides its own estimates of  $V_{\rm air}$ , which are derived solely from single-sensor observations by the CPR (Sato et al. 2025). Both the method implemented in CPR\_CLP and the CPR-DPR combined approach presented in this study are based on the same fundamental concept of calculating  $V_{\rm air}$  by subtracting the  $V_t$  from the  $V_d$ . However, the two methods differ in how  $V_t$  is determined: the CPR\_CLP estimates it from a particle size distribution (PSD) inferred using only the CPR-measured  $Z_w$  and  $V_d$ , whereas the present method uses the PSD derived from the 2A.DPR algorithm. Because of this difference in the underlying PSDs and scattering database, discrepancies between the two  $V_{\rm air}$  estimates are expected. Therefore, in this section, we compare the two  $V_{\rm air}$  estimates to assess the consistency and reliability of the retrieved  $V_{\rm air}$ , as well as the implicit assumptions regarding  $V_t$  and PSD within each algorithm. Since the CPR\_CLP product does not provide the quantity corresponding to  $V_t$ , the comparison focuses solely on  $V_{\rm air}$ .

Figure 11 shows histograms of  $V_{\rm air}$  for deep stratiform and deep convective precipitation, separately for the temperature ranges corresponding to snow (-10 °C < T < 0 °C) and rain (T > 4 °C). Table 2 summarizes the mean and standard deviation of the  $V_{\rm air}$  histograms shown in Fig. 11 for each retrieval method. Overall, the histograms exhibit smaller standard deviation in stratiform cases (0.5–0.9 m s-1) and larger standard deviation in convective cases (0.9–1.6 m s-1), likely associated with turbulent motions.

For the snow layers (Figs. 11a and 11b), weak upward motions appear on average in both types (0.3–0.6 m s-1), likely associated with latent heat release during ice particle growth. On the other hand, some discrepancies exist between the two  $V_{\rm air}$  estimates, probably due to the radar frequency and assumptions about ice particle properties. The CPR\_CLP algorithm accounts for scattering from ice particles with various shapes and orientations, whereas the DPR algorithm assumes simple spherical particles with a fixed bulk density of 0.10–0.13 g cm-3. The peak of the  $V_{\rm air}$  distribution from the DPR-based method is about 0.3 m s-1 higher than that from CPR\_CLP, suggesting a positive bias. This difference may indicate that the DPR-based PSD retrieval does not fully capture the contribution from smaller particles with slow  $V_t$  compared with the W-band CPR-based retrieval. On the other hand, for convective precipitation, the DPR-derived  $V_{\rm air}$  shows a larger proportion of downward motions (< -1 m s-1) than the CPR\_CLP  $V_{\rm air}$ . This is likely because convective cases include more dense particles, such as graupel or hail, with densities exceeding 0.3 g cm-3, which are not represented in the current 2A.DPR algorithm.

For the rain layers (Figs. 11c and 11d), the DPR-based method and CPR\_CLP show more similar  $V_{\rm air}$  histograms than in the snow cases. For stratiform precipitation, the mean  $V_{\rm air}$  is close to 0 m s-1, whereas for convective precipitation, it shows stronger upward motion of about 1 m s-1 with a larger variance. The two histograms agree well for strong upward motion ( $V_{\rm air}$  > 1 m s-1); however, for  $V_{\rm air}$  < 1 m s-1, the CPR\_CLP  $V_{\rm air}$  shows a distinct peak around 0 m s-1, while the DPR-based  $V_{\rm air}$  exhibits a smoother distribution. Despite such several differences, both methods yield comparable  $V_{\rm air}$  histograms overall.

While further validation is still needed, the approach presented in this study provides useful reference information for evaluating the validity of vertical air motion retrieval, which can otherwise only be obtained by specific ground-based vertically pointing radars where direct observational data are extremely limited.

Figure 1: Histograms  $V_{air}$  calculated using the CPR-DPR combined method proposed in this study, and that using CPR\_CLP (a, b) for the temperature range from  $-10^{\circ}$ C to  $0^{\circ}$ C (snow) and (c, d) for the temperature above  $4^{\circ}$ C (rain). (a, c) deep stratiform and (b, d) deep convective types. Each histogram is normalized by the total number of samples for each precipitation type.

Table 2: The mean and standard deviation of each  $V_{air}$  histogram shown in Fig. 11.

| Indicator          | V air retrieval method | Snow $(-10^{\circ}\text{C} < T < 0^{\circ}\text{C})$ |               | Rain (T > 4°C) |               |
|--------------------|-----------------------------------|------------------------------------------------------|---------------|----------------|---------------|
|                    |                                   | a) Stratiform                                        | b) Convective | c) Stratiform  | d) Convective |
| Mean               | DPR-based                         | 0.561                                                | 0.356         | 0.279          | 1.112         |
|                    | CPR_CLP                           | 0.302                                                | 0.621         | 0.375          | 1.126         |
| Standard deviation | DPR-based                         | 0.638                                                | 1.366         | 0.874          | 1.631         |
|                    | CPR_CLP                           | 0.487                                                | 0.945         | 0.738          | 1.198         |

Furthermore, we have added the following discussion to the summary in Section 4:

**Line 646-652**

In addition, we compared the  $V_{air}$  estimated using the DPR-derived PSD with that estimated solely from CPR measurements in the CPR\_CLP product. The two estimates show consistent characteristics, exhibiting smaller variability with values concentrated around 0 m s-1 for stratiform cases and larger variability for convective cases, likely associated with turbulent motions. On the other hand, a systematic bias is found between the two in the snow layer, which can be attributed to differences in the microphysical assumptions and observational characteristics due to radar frequency. Such information is expected to be valuable for improving both algorithms and for providing reference data to validate vertical velocity retrievals, which are otherwise extremely limited in direct observations.

2. In the paper, V\_t and V\_air are shown only for rain layers (Fig. 10). Why not add V\_t and V\_air for solid precipitation too (Fig. 9)? The DPR Dm might not be as reliable for snow as for rain because the KuPR is not

sensitive enough to small frozen hydrometeors, but a comparison with the CPR\_CLP product would be worth studying for solid precipitation as well.

We appreciate the reviewer's suggestion. As you pointed out, including the results for ice-phase precipitation allows for a more comprehensive discussion, including the limitations of DSD retrievals by DPR in snow. Therefore, we attempted to estimate  $V_{\rm air}$  based on DPR-derived DSDs for snow in the temperature range from  $-10^{\circ}$ C to  $0^{\circ}$ C. The results and related discussion have been incorporated into the revised manuscript at Line 568–608, as described in our response to the previous comment.

In addition, we have added the following description to Section 2 regarding the assumed snow particle size distribution and terminal fall velocity used in the DPR for the calculation of  $V_t$ :

**Line 237-257**

For rain layers,  $v_t$  was computed using the empirical relationship proposed in Atlas and Ulbrich (1977), with a correction factor for air density, as given:

$$v_t(D) = -3.78D^{0.67} \cdot c(\rho), \tag{5}$$

$$c(\rho) = \sqrt{\frac{\rho_0}{\rho}} = \sqrt{\frac{\rho_0 RT}{p}}.$$
 (6)

Here, the unit of D is millimeters,  $\rho$  denotes the ambient air density,  $\rho_0$  is the standard air density (set to 1.225 kg m-3), R is the specific gas constant for dry air (287 J kg-1 K-1), and p and T represent pressure and temperature obtained from auxiliary data. The backscattering cross-section  $\sigma_b$  was derived from Mie scattering calculations for spherical raindrops at W-band frequency.

For snow,  $\sigma_b$  and  $v_t$  was calculated in the same manner as in the 2A.DPR algorithm, assuming homogeneous spherical particles with a density of 0.10–0.13 g cm-3 and a melted-equivalent diameter following the particle size distribution given by Eq. (3). The terminal fall velocity of snow was calculated following Magono and Nakamura (1965) as follows:

$$v_t(D_s) = -8.8(0.1D_s\rho_s)^{0.5} \cdot c(\rho), \tag{7}$$

where  $D_s$  is the unmelted snow particle diameter in mm, and  $\rho_s$  is the density of snow particles in g cm-3. On the other hand, ice particles can take various shapes, sizes, and densities, such as those of snow, graupel, and hail. Because  $\sigma_b$  and  $v_t$  vary depending on these parameters, the assumptions made for snow in this study are often not valid. Although it would be ideal to account for more realistic and complex scattering and fall characteristics of ice particles (Kuo et al. 2016; Ori et al. 2021), considering such diversity is challenging because the CPR observes only in the nadir direction and therefore cannot provide information on particle asymmetry. This contrasts with ground-based dual-polarization radars, which observe the hydrometeor from the side, where particle asymmetry is more evident and can provide additional information. In addition, such information on particle diversity cannot be inferred from the current version of the 2A.DPR algorithm and is therefore left for future work.

**Specific points -----**

**3. I. 55: and The TRMM -> and the TRMM**

We have modified the sentence as follows:

Line 55

The Tropical Rainfall Measuring Mission (TRMM) satellite was launched in 1997, and it carried the world's first spaceborne Ku-band (13.8 GHz) radar, Precipitation Radar (PR)

**4. I. 73: weaker hydrometeors -> smaller hydrometeors**

We have revised the manuscript as suggested.

5. I.135: are more frequently appear -> more frequently appear

We have revised the manuscript as suggested.

6. II. 144-146: I am puzzled by the claim that "only data from the HS mode are used". The KaHS mode has been reassigned to match the outer KuPR swath to complement the inner KaMS swath since the scan pattern was changed in May 2018. As far as I can tell from Figs. 2 and 3, the whole DPR swath (that is, both MS and HS modes) seems to be analysed in this work.

As pointed out, KaHS is used in Figures 2 and 3. Our intention was to indicate that only KaHS was used in the statistical analysis presented in Section 3.2 and thereafter. We have revised the text to make this point clear as follows:

Line 151-153

In this study, only data from the HS mode are used in the statistical analysis from Section 3.2 onward,

7. I. 153: which calculated -> which were calculated

We have revised the manuscript as suggested.

8. I.156: "equatorial" region may be better replaced by low- and mid-latitude regions. 65S-N is much wider than the equatorial region.

We have revised the manuscript as suggested.

9. l. 170: samplings -> samples

We have revised the manuscript as suggested.

10. Figure 4c/d: Why are the KuPR and KaPR CFEDs sharply (presumably artificially) cut off above a certain level, with a temperature threshold around -42C for Ku and -35C for Ka?

The DPR has lower sensitivity than the CPR, and therefore, in the upper regions, there are many cases where echoes are detected by the CPR but not by the DPR. Consequently, the sample size in the colder temperature range in Figs. 4c and 4d is considerably smaller than that for the CPR. To avoid misinterpretation, temperature ranges with fewer than 1000 samples are not shown. The following note has been added to clarify this point.

Lines 314-316

Because the sensitivity of the DPR is lower than that of the CPR, the number of samples in the colder temperature range in Figs. 4c and 4d is considerably smaller than that for the CPR. To avoid misinterpretation, temperature ranges with fewer than 1000 samples are not shown.

11. I. 303: "However" does not really fit the context here. Please try "On the other hand" or "In contrast" instead. We have revised the manuscript as suggested.

12. Figure 6: The joint histogram of Ku- and W-band Z reminds me of Fig. 8 of Stephens and Wood (2007, DOI: 10.1175/MWR3321.1). They showed CFADs separated for different cloud types, which also bears resemblance to the present work. I just thought this might be worth a brief discussion.

We appreciate the reviewer's suggestion. The classification based on precipitation and cloud-top height by Stephens and Wood (2007, hereafter SW07), as well as their subsequent analysis using joint histograms, is closely related to our approach and supports our interpretation. Therefore, we have added acknowledgments to SW07 and the related study by Masunaga et al. (2005), along with a brief discussion of the similarity to SW07, in Section 3.3 of the revised manuscript as follows:

**Line 399-410**

The cloud top height (CTH) and precipitation top height (PTH) are key variables that characterize the developmental stage of precipitation systems (Masunaga et al., 2005, hereafter M05; Stephens and Wood 2007, hereafter SW07; Takahashi and Luo, 2014; Kikuchi and Suzuki, 2018). M05 first categorized precipitation systems using CTH–PTH joint histograms constructed by deriving the PTH from the 18-dBZ echo top observed by the TRMM PR and the CTH from the 11-µm brightness temperature observed by the Visible Infrared Scanner (VIRS) onboard TRMM. SW07 improved upon this approach by incorporating millimeter-wavelength radar observations, which allowed them to better represent multilayer cloud structures, whereas VIRS observations can capture only the uppermost cloud layer. Following these studies, this work assumes that the ETH retrieved from CPR corresponds to the CTH, and that from KuPR corresponds to the PTH. The joint histograms of temperature at CTH and temperature at PTH for stratiform and convective precipitation determined by 2A.DPR algorithm, respectively, are shown in Fig. 6. The histograms are reminiscent of the histograms presented by M05 (their Fig. 1) and SW07 (their Fig. 8).

**Line 464-468**

SW07 has classified cloud types using a similar approach based on ground-based Ka-band radar observations and presented comparable histograms of Z as a function of height (their Fig. 10). Although the attenuation conditions differ—since their observations are made from the ground upward, whereas the present study is based on spaceborne downward-looking radar measurements—the vertical profiles of  $Z_W$  in each category (Fig. 7a–d) exhibit similar characteristics. This study extends their findings by introducing an additional perspective through the use of  $V_d$ .

13. I.375: "Intense" deep convection (DC-I) could be misleading, given that higher echo tops do not necessarily guarantee more intense convection (e.g., Hamada et al., 2015, DOI: 10.1038/ncomms7213). Something like Tall deep convection (DC-T) may be a safer alternative.

We thank the reviewer for this careful comment. As pointed out, a high precipitation top height does not necessarily indicate strong precipitation. As suggested, we have changed the name from *Intense deep convective (DC-I)* to *Tall deep convective (DC-T)*, and replaced the corresponding terminology throughout the manuscript.

14. I. 394: is generally low -> are generally low We have revised the manuscript as suggested.

15. I. 415: Here again, "However" may be better rephrased by "By contrast" etc.

We have revised the manuscript as suggested.

---

## Author Response (AR2)

**Response to editor report**

**Public justification (visible to the public if the article is accepted and published)**:
The authors created EarthCARE-GPM (Global Precipitation Measurement core observatory) coincident data set and examined reliability of the EarthCARE CPR measured Ze and Vd values. They also derived the vertical air motion by the combined method of CPR and GPM. The retrieved vertical air motion was then compared with that in the JAXA standard CPR-CLP product and found reasonable agreement. The manuscript was revised according to the reviewers comments and the paper is now accepted with technical correction.

The authors sincerely thank the editor for carefully reading the manuscript and for evaluating our work as worthy of publication in AMT. In accordance with your comments, we have revised the manuscript as well as the short summary.

(1) The authors used Vt and vt to denote reflectivity weighted terminal velocity and fall velocity(not reflectivity weighted values), respectively. The authors are requested to use other name for reflectivity weighted terminal velocity, e.g., VTz instead of Vt to avoid misunderstanding,

Thank you for the suggestion. As you advised, we revised the entire manuscript to use $V_{tz}$ instead of $V_t$, including the labels shown in Figure 10.

(2) In Short summary, please use Global Precipitation Measurement core observatory (GPM) for readers who are not familiar with GPM.

We have revised the short summary as follows:

Using coincident observations from the EarthCARE Cloud Profiling Radar with Doppler velocity measurement capability and the Dual-Frequency Precipitation Radar on the Global Precipitation Measurement, vertical motions in stratiform and convective precipitation systems are examined, providing insights into the dynamical and microphysical processes inside deep clouds. This enables a more comprehensive understanding of hydrometeor fall speeds and vertical air motions in precipitation systems.

In addition, although these were not specifically pointed out, we made the following two updates to the manuscript:

1. The affiliation of one of the co-authors has changed as of November, so we updated it accordingly.

   Affiliation of F. Joseph Turk:
   → Joint Institute for Regional Earth System Science and Engineering, University of California, Los Angeles, CA, USA

2. The statement regarding data availability was missing, so we have added it as shown below.

   Line 671–674

*Data availability*

EarthCARE-GPM coincidence dataset is publicly available on the JAXA website under the Data DOI, https://doi.org/10.57746/EO.01ka7xakvwj6pcthxkvgt0vr0y. All EarthCARE products (JAXA, 2024a; JAXA, 2024b) and GPM products (JAXA, 2014) used in this study can be downloaded from the JAXA G-Portal.